

# Multi-year gradient measurements of sea spray fluxes over the Baltic Sea and the North Atlantic Ocean

Piotr Markuszewski[1,2,3], E. Douglas Nilsson[2,3], Julika Zinke[3,4], E. Monica Mårtensson[5], Matthew Salter[3,4], Przemysław Makuch[1], Małgorzata Kitowska[1], Iwona Niedźwiecka-Wróbel[1], Violetta Drozdowska[1], Dominik Lis[6], Tomasz Petelski[1], Luca Ferrero[7], Jacek Piskozub[1]

[1]Physical Oceanography Department, Institute of Oceanology, Polish Academy of Sciences, Sopot, 81-712, Poland
[2]Department of Environmental Science, Stockholm University, Stockholm, 10691, Sweden
[3]Bolin Centre for Climate Research, Stockholm University, Stockholm, 10691, Sweden
[4]Baltic Sea Centre, Stockholm University, 10691 Stockholm, Sweden
[5]Department of Earth Sciences, Uppsala University, Uppsala, 75105, Sweden
[6]Marine Physics Department, Institute of Oceanology, Polish Academy of Sciences, Sopot, 81-712, Poland
[7]Luca Ferrero, GEMMA and POLARIS Centre, Department of Earth and Environmental Sciences, University of Milano-Bicocca, 20126 Milano, Italy

*Correspondence to*: Piotr Markuszewski (pmarkusz@iopan.pl)

**Abstract.** Ship-based measurements of sea spray aerosol (SSA) gradient fluxes in the size range 0.5-47 µm diameter were conducted between 2009 - 2017 in both the Baltic Sea and the North Atlantic Ocean. Measured total SSA fluxes varied between $8.9 \cdot 10^3 \pm 6.8 \cdot 10^5$ m$^{-2}$ s$^{-1}$ for the Baltic Sea, and $1.0 \cdot 10^4 \pm 10^5$ m$^{-2}$ s$^{-1}$. for the Atlantic Ocean. The analysis uncovered a significant decrease (by a factor of 2.2 in wind speed range 10.5 m s$^{-1}$ - 14.5 m s$^{-1}$) in SSA fluxes with chlorophyll-*a* (Chl-a) concentration higher than 3.5 mg m$^{-3}$ in the Baltic Sea area. We found statistically significant correlations for both regions of interest between SSA fluxes and various environmental factors including wind speed, wind acceleration, wave age, significant wave height, and wave Reynolds number. Using these factors, we developed separate parameterizations and compared them with previous studies. Additionally, in both measurement regions we observed weak correlations between SSA fluxes and air and water temperature, as well as atmospheric stability. Comparing the Baltic Sea and North Atlantic, we noted distinct emission behaviours, with higher emissions in the Baltic Sea at low wave age values compared to the Atlantic Ocean. This study represents the first comparative analysis of SSA flux measurements using the same methodology in these contrasting marine environments.

## 1 Introduction

### 1.1 Sea spray properties and source processes

Sea and ocean surface is one of the two largest natural aerosols source (Seinfeld and Pandis, 2006). The production of marine primary aerosol, often referred to as sea spray aerosol (SSA), remains a poorly understood aspect of the marine boundary layer (Quinn et al., 2017). The multitude of factors influencing SSA emission and subsequent turbulent transport, contributes to



significant uncertainty in global estimates of SSA production (Tsigaridis et al., 2013). This uncertainty derives from several not well understood processes: the turbulence in the atmospheric surface layer, the turbulence in the surface ocean and their interaction, therefore including the formation of waves, wave breaking and entrainment processes. It also encompasses the

formation of bubble clouds from the entrained air, and the formation of water droplets, from the bubbles. A substantial gap of knowledge is present within the aforementioned multiple processes. However, accurate estimates of SSA flux and precise parameterization of its vertical transport through turbulent diffusion are essential across various branches of geoscience (de Leeuw et al., 2011).

The release of SSA from the sea surface is initiated by wind-induced wave-breaking (Nilsson et al., 2001, 2021; Yang et al.,

2019; Zinke et al., 2024). Following wave breaking, air bubbles are entrained into the water column, ascend to the surface, and burst. Traditionally, two types of aerosol droplets resulting from this bubble bursting are identified, namely film and jet drops (Spiel, 1998, Wolf et al., 1987). In high-wind conditions, spume drops are also emitted from the sea surface (Mehta et al., 2019). Subsequently, turbulent diffusion intercepts all three types of droplets, transporting them into the atmosphere within the marine boundary layer.

SSA plays a crucial role in atmospheric chemistry, acting as a primary source of both inorganic and organic aerosol (Cochran, et al., 2017); moreover, SSA can act as a sink for semi-volatile gases (biogenic or anthropogenic) and can prevent secondary aerosol generation, including new particle formation (Carslaw et al., 2010). SSA also affects the atmospheric load of cloud condensation nuclei (CCN) and cloud physics (Xu et al., 2022). The chemical composition of SSA and their degree of internal mixing are other areas of research with large gaps in, and these are probably closely connected the formation of bubbles. Since

it is known that while the bubble plume rise towards the sea surface they have a great affinity to collect all sorts of surfactants, first enriched in the bubbles, and as the bubbles burst, enriched also in the SSA aerosol. This applies to natural organic surfactants (Facchini et al., 2008), as well as anthropogenic surface active substances (Oppo et al., 1999; Johansson et al., 2019).

Once acting as CCN, SSA may interfere with water phase chemistry in the cloud droplets. Their significance extends to the

global radiative balance through both direct aerosol effects (Bates et al., 2006; Mulcachy et al., 2008; Vaishya et al., 2012; Rap et al., 2013) and indirect effects by acting as CCN (O'Dowd et al., 1999; Andrae and Rosenfeld, 2008), and as such must be considered in regional and global climate modelling (Partanen et al., 2014).

SSA, apart from its influence on climate, has several other significant aspects. One important aspect is its role in the transport of pollutants from the sea to the air like perfluoroalkyl acids (Sha et al., 2020, 2021), or microplastics (Alen et al. 2020, Ferrero

et al., 2022).

Moreover, composition of SSA vary with the seasons and biological activity (Parent et al., 2023). It has been shown to contain a significant organic component (Cavalli et al., 2004; Facchini et al., 2008), which has implications for its hygroscopicity and ice nucleation activity (Darr et al., 2018). Furthermore, SSA has been demonstrated to be geochemically important over geologic timeframes due to the presence of trace elements and metal pollution in aerosols derived from sea spray (Marx et al.,

65    2014).



Despite recognizing the significance of incorporating SSA into the aerosol budget of the marine boundary layer, the development of parameterizations for this aerosol source remains highly challenging and a wide array of parameterizations have been proposed, drawing from both laboratory studies (Monahan et al., 1982, 1994; Mårtensson et al., 2003; Keene et al., 2007; Tyree et al., 2007; Long et al., 2011; Salter et al., 2015) and field studies (Nilsson et al., 2001; Geever et al., 2005; Norris et al., 2008, 2012; Yang et al., 2019; Nilsson et al., 2021; Zinke et al., 2024), here limiting the list only to such studies where SSA emission fluxes where directly observed using the Eddy Covariance (EC) method.

Due to the fact that sea surface is impacted by synergies of different factors cause high uncertainty in SSA emission parameterization. For instances, there are indications that sea surface temperature (SST) can influence the sea spray flux through changes in surface tension and kinematic viscosity (i.e. Bowyer et al., 1990; Mårtensson et al., 2003; Hultin et al., 2011; Zabori et al., 2012, Foresteri et al., 2018) through changes in the bubble spectra (Salter et al., 2014; Zinke et al., 2022). The exact processes in this link are not understood, but bubble coalescence is the most likely candidate (Ribeiro and Meiwes, 2006).

## 1.2 Gradient method for aerosol fluxes

The studies related to the use of vertical aerosol profiles in the boundary layer have a very long history. The first vertical aerosol profile was measured by John Aitken (Aitken, 1890; Podzimek, 1989). Since these works, a great deal of progress has been made. The understanding of the aerosol profiles in the atmospheric boundary layer increased thanks to the advances in measurement techniques, like tower measurements or lidar remote sensing.

Recent technological developments of unmanned platforms (e.g. tethered balloons, Ferrero et al. 2014, or drones, Chiliński et al. 2017) as well as miniaturized, low-cost sensors have led to further progress in vertical aerosol profiling. This led to the possibility to use the gradient method (GM) for marine aerosol fluxes measurement. In this respect, this work is the continuation of research started by Petelski (2003), in which the GM was applied for the first time evaluating aerosol profiles measured on board of a ship. We present the theory of gradient flux calculation in Section A in the Appendix.

The GM has been also successfully applied to derive SSA generation functions from the North Atlantic (Petelski & Piskozub, 2006; Andreas, 2007), Pacific Ocean (Savelyev et al. 2014) and well as in the Baltic Sea (Petelski et al., 2014; Markuszewski et al., 2016, 2020).

The SSA source function from North Atlantic waters using GM is calculated by Petelski and Piskozub (2006) (and was later improved by Andreas 2007). Later, the first generation function of sea spray for the Baltic Sea was estimated using this method (Petelski et al., 2014).

The GM was further successfully applied for ship based measurements in the Pacific Ocean by Savelyev et al. (2014). The so derived SSA fluxes were compared with the deposition method and parameterized with surface brightness temperature.

The case studies of gradient aerosol fluxes obtained from two ship based campaigns in the Baltic Sea cruises are presented by Markuszewski et al. (2017). The results showed the influence of wave properties calculated based on wave height and peak frequency on sea spray fluxes.





Another case study was presented by Markuszewski et al. (2020), where gradient sea spray fluxes were compared with
underwater sound pressure level in relation to development of wave state. One of the main results of that study was showing
the impact of wind history and wave development (in two regimes: developing and developed) on hydroacoustic bubbles noise
(represented by power spectrum density of noise) and SSA fluxes.

### 1.3 Flux parameterisation

In order to present sea spray dependence on different factors, so-called Sea Spray Generation Functions (SSGF) (or Source
Functions) are used. The first SSGF was introduced by Monahan et al. (1982), where the laboratory experiment of whitecap
simulation was used. Since then, a lot of different approaches have been given in the literature (Lewis and Schwartz, (2004),
de Leeuw et al., (2011), Grythe et al., 2014, Veron (2015)).

This approach is commonly used and is represented as a combination of aerosol size distribution and driving parameterisation:

$$\frac{\mathrm{d}F(D_p, \alpha, \beta, \dots)}{\mathrm{d}D_p} = g(D_p)h(\alpha, \beta, \dots). \tag{1}$$

Where $F$ stand for aerosol flux, $D_p$ particle diameter, $g$ and $h$ are separate source functions, $\alpha$, $\beta$,.. represents parameters that
could affect the SSGF such as: wind speed, sea surface temperature, etc. We discuss these factors in the following sections.
One of the first SSGF was introduced by Monahan et al. (1982; 1986), where laboratory experiments of artificial breaking
waves were made in a water tank. All parameterisations that are based on laboratory experiments need a method to apply these
on the real atmosphere-ocean interface and atmospheric surface layer. A common approach is the whitecap surface on the
ocean scaled to the water surface with bubbles in the laboratory tank (e.g. Mårtensson et al., 2003; Tyree et al., 2007; Fuentes
et al., 2010) or the decay time scale of white caps (Monahan et al. (1982; 1986). A later approach is the scale the air entrainment
in a laboratory tank to the air entrainment over the real ocean (Salter et al., 2015; Deike et al., 2022). Since direct measurements
of sea spray fluxes using EC became available starting with Nilsson et al. (2001), some laboratory based parameterisations
have been constrained by in situ EC fluxes (e.g. Mårtensson et al., 2003), evaluated by independent EC fluxes (Norris et al.,
2008; Yang et al., 2019; Nilsson et al., 2021; Zinke et al., 2024), or used to derive new source parameterisations from in situ
EC fluxes (Norris et al., 2012; Zinke et al., 2024). This has improved the overall quality of SSGFs and decreased the
discrepancies between them. However, the EC flux method is in practise only useful for particle diameters <1 µm. The aerosol
instruments count the number of particles and the super micrometre particles are simply too few. Therefore, the gradient
method can make an important contribution especially for super micrometre particles.

### 1.3.1 Wind speed and wind history

Wind speed is the main parameter that greatly influences sea spray emission. Wind speed: 1) creates the drag between
atmosphere and ocean that builds up the waves until they break, and 2) generates turbulent diffusion transport that is
responsible for vertical transport of ejected aerosols from ocean surface through the atmospheric surface layer to the rest of
the troposphere. Wind speed dependence are included in most SSGFs. For parameterisations based on laboratory experiments



the wind dependency is usually part of the equation that relate the tank SSA production to the real ocean surface, as mentioned above derived from whitecap surface relationships, whitecap decay time scale, or air entrainment. The whitecap fraction ($W$) usually is proportional to the wind speed with a power law relationship where power exponent $\lambda$ varies from 2 to 4 (e.g. Monahan and O'Murtaigh, 1980; Callaghan et al., 2008), $\lambda$=3.41 the most common (Monahan and O'Murtaigh, 1986, Hansson and Phillips, 1999; Long et al., 2011).

SSGFs based on EC flux measurements have resulted in exponential functions of $U_{10}$ (Nilsson et al., 2001, Geever et al., 2005; Norris et al., 2012; Zinke et al., 2024). Gradient flux data has resulted in SSGFs with a square wind dependency ($U_{10}^2$) (Petelski, 2005) or exponential functions (Petelski and Piskozub, (2006), Andreas (2007).

Another parameter connected with the development of the sea surface is wind acceleration (or wind history parameter). This parameter can be defined the same as Hanson and Phillips (1999) or Calaghan (2008) as wind acceleration:

$$a_U = \frac{\overline{\Delta U_{10}}}{\Delta t}.$$
(2)

Here $\overline{\Delta U_{10}}$ represents averaged wind speed over wind acceleration time (in this work we assumed 2 hours) and $\Delta t$ is the time step interval between flux measurements.

**1.3.2 Wave state**

Apart from wind speed another factor that can affect SSA emissions is wave state. Massel (2007) used dimensional analysis
to link aerosol emission with significant wave height and peak frequency. According to Norris et al (2012), another parameter influencing SSA emissions is the mean wave slope (which combines significant wave height with mean wave period). Later on, Norris et al. (2013), Ovadnevaite et al., (2014), and recently Yang et al., (2019) and Zinke et al., (2024) show a linear correlation between EC sea spray flux and the wave state-dependent Reynolds number ($ReH_w$) (which includes friction velocity, significant wave height, and water viscosity – which is weakly related to water temperature).

In order to understand the relationship between aerosol emission and wind waves, it is essential to describe the mechanisms of wind waves generation. Wind drag on the sea surface is responsible for the initiation of the generation of wind waves. The turbulent kinetic energy transport from atmosphere causes pulsation of pressure, which generates normal tension to the sea surface. The resonance between these pulsations and the random response of the free sea surface is called a Phillips generation mechanism (Phillips, 1957). Similarly, while waves are growing the shear stress becomes increasingly important in the
development of the wave field. Finally, the shear stress causes even more an increase of waves energy. This type of generation process is called the Miles mechanism (Miles, 1957). Profound analyses of known models of waves generation mechanisms are given, among others by Phillips (1977) or Massel (1996).

One of the parameters that gives information about described mechanisms is the so called wave age $w_a$. The dimensionless wave age is defined as the ratio between wave phase velocity $c_p$ and wind speed (Massel, 2010).:

$$w_a = \frac{c_p}{U_{10}}$$
(3)

where the wave phase velocity is defined as (deep water assumption):





$$c_p = \frac{g}{2\pi} t_p \tag{4}$$

Where g is the gravitational acceleration, and $t_p$ is wave period. The wave age brings information on the state of the wave field development. If the wave phase is lower than the wind speed, the wave field is developing and the waves are 'young'. In the

reverse situation where the velocity of waves is higher, wave field is developed and waves are "old".

The original application of the Reynolds number to describe the wave state and wave breaking was given by Zhao and Toba, (2001):

$$ReH_w = \frac{u_* H_s}{\nu}. \tag{5}$$

Where, $u_*$ is a friction velocity, for the length-scale parameter that is always part of the Reynolds number we here use $H_s$ and

$\nu$ represents kinematic viscosity in denominator. The first application of wave Reynolds number in parameterisation of gas transfer velocity was proposed by Woolf (2005), in dependence on water viscosity (notation $ReH_w$). Ovadnevaite et al., (2014) were the first to apply the wave Reynolds number to parameterise SSA fluxes it in parametrisation of sea spray fluxes. Recently this parameter was also investigated by Yang et al., (2019), and Zinke et al., (2024).

### 1.3.3  Seawater temperature

Another parameter influencing the sea spray emission is water temperature ($T_w$). This effect was first discovered in laboratory experiments by Bowyer et al. (1990) using the same wave breaking tank as Monahan et al. (1982). It was extended to sub micrometric particles by Mårtensson et al., (2003). For sub micrometric particles, there is a near consensus with a decline in particle production with increasing temperature, especially  below15˚C (Mårtensson et al., 2003; Hultin et al., 2011;  Salter et al., 2014, 2015; Zinke et al., 2022; Sellegri et al, 2023, both with artificial sea water and in situ with local sea water). Super

micrometre SSA may have the opposite temperature trend (Bowyer et al., 1990; Jaeglé et al., 2011; Drod et al., 2018). The temperature trend may also be modified or disappear if wall effects are allowed to influence how the bubble spectra evolve. Salter et al. (2015) presented an updated temperature-based SSGF using state of the art measurements of sea spray production in a sea spray simulation tank. Extension of this research was recently presented by Zinke et al., (2022) where in a similar experiment the influence of salinity and temperature was investigated.

### 1.3.4. Atmospheric stability

There are very few attempts of combining sea spray emission with water and air temperature difference ($T_d = T_w - T_a$), sometimes called as bulk atmospheric stability. SSA emission may be represent indirectly by $W$ accordingly to Monahan and O'Muircheartaigh, (1985) and Monahan (1986). According to these papers the relation of $W$ should be proportional to the value $T_d$. This means that, for an unstable atmosphere ($T_d > 0$ºC), an higher whitecapping is expected than for a stable one ($T_d < 0$ ºC).

However, the effect seems to be not easy to capture, that is why there were difficulties with reproduction of these results by other researchers. For instance, Stramska and Petelski (2003) reported no correlations between whitecap coverage and atmospheric stability.



### 1.3.5. Marine biological activity

Since the last review of the topic (Gantt and Meskhidze, 2013) there is still lack of unequivocal consensus how the marine
biological activity influences SSA emissions. Recent findings given by Bates et al., 2020 supports previous studies (Gantt et
al., 2009; Long et al., 2011; Ault et al., 2013) in describing that the role of organic matter or marine biological blooms have a
minor impact on marine primary emitted aerosol. Keene et al., (2007), Facchini et al., (2008), Quinn et al., (2014), Alpert et
al., (2015), Long et al., (2014), or more recent Christiansen et al., (2019) and Sellegri et al., (2023) suggest that presence of
surfactants (surface active agents) may modulate the SSA emission; the surfactant amount can be represented as total organic
carbon or Chl-a (chlorophyll-*a* concentration). Several studies (e.g. Keene et al., 2007; Facchini et al., 2008)  showed that sea
salt dominates the super micrometre sea spray. Below that, the importance of organic sea spray increase, until it reaches about
80 % at 0.1 μm $D_p$.  Based on this, organic sea spray should have a limited effect on the current study. Nilsson et al., (2021)
demonstrated with EC flux measurements that the presence of organic sea spray from microbiological activity changed the
wind dependency of the sea spray aerosol emissions.

### 1.4 Objectives of this study

In this study, we analysed a comprehensive dataset of measured sea spray fluxes obtained aboard a research vessel in open sea
conditions within the Baltic Sea and North Atlantic Ocean regions. The sea spray fluxes were obtained using the gradient
method (Petelski et al., 2003). Our analysis of these data addresses the following research objectives:

1.  What is the impact of selected meteorological and oceanographic parameters (wind speed $U_{10}$, wind acceleration $a_U$,
wave age $w_a$, significant wave height $H_S$, wave Reynolds number $ReH_w$, sea surface temperature $T_w$, air temperature
        $T_a$, atmospheric stability $T_d$, and chlorophyll-*a* concentration Chl-a) on SSA fluxes?
2.  Can we identify relations that are useful as source parameterisations, or which may improve existing
        parameterisations?
3.  How do our results relate to previous relevant source parameterisations?
4.  How do SSA fluxes from the Baltic Sea compare to fluxes from the Atlantic Ocean (in terms of SSA fluxes magnitude
        as well as the above impact of selected parameters)?

### 3 Measurements, methods and data

### 3.1 Measurement platform and study areas

#### 3.1.1 RV *Oceania*

All measurements were carried out on board the RV *Oceania*. The ship is owned by the Institute of Oceanology Polish
Academy of Sciences (https://old.iopan.pl/oceania.php). The ship is a sailing vessel (length: 48.9 m, width: 9 m, draught: 3.9
m). Owing to this fact, it is well suited for all atmospheric observations due to the main body of the ship being near the water



surface. When using a ship as platform for flux studies, the mean air flow is tilted upward by the ship (e.g. Landwehr et al, 2015; Losi et al., 2023). The low profile of the RV *Oceania* implies that the flow distortion is small. The ship has three masts

(32 m tall each).

Measurements were carried out in two different marine environments: in the southern Baltic Sea area (Baltic Proper) and in the northern Atlantic Ocean (Norwegian Sea, Greenland Sea). The locations of the measurements are presented in figure 1. The scientific equipment used for this study was installed on the balcony on the foremast at 10 m above sea level (the meteorological probe), and on special lift in the right side of the ship moving in levels from 8 m to 20 m, see below.

### 230 3.1.2 The Baltic Proper

The Baltic Sea is one of the largest closed brackish seas. Fresh ocean deep water inflows are rare (Rak, 2016). The wave fetch is much shorter than in the open ocean, and depends on the wind direction and the upwind trajectories. That is why in this environment younger waves dominate (Lepparanta and Myrberg, 2009). As such, the Baltic Sea is characterised by different wave conditions (e.g. younger waves) compared to the open ocean.

In this study 224 hours of data gathered between 2011 to 2017 in twelve research cruises were used. The cruises were conducted mostly during late autumn (October/November) and winter (January/February). Those periods were chosen due to the increased occurrence of high wind speeds. Main measurement stations from Baltic measurements are presented in Fig. 1a. Dates and positions of measurements stations are shown in the Appendix A Table A1.

### 3.1.3 North Atlantic Ocean

Measurements in the North Atlantic Ocean were conducted during summer. Aerosol measurements are included in the larger project, so called AREX (Arctic Experiment, Walczowski et al. 2017). The AREX cruises are annually organized three month long Arctic research cruises on board RV *Oceania* since 1986.

During these campaigns, multidisciplinary marine observations were conducted. The first part of the cruises was devoted to hydrological measurements. Aerosol concentration gradient measurements were carried out if meteorological and technical

conditions were appropriate during each station (i.e. the station had to last more than one hour, without occurrence of rain or





fog). Despite difficult conditions, we succeeded to gather in total, 56 hours of measurements between 2009 – 2017 from six cruises. Aerosol measurement points during AREX campaign are presented in Fig. 1b.

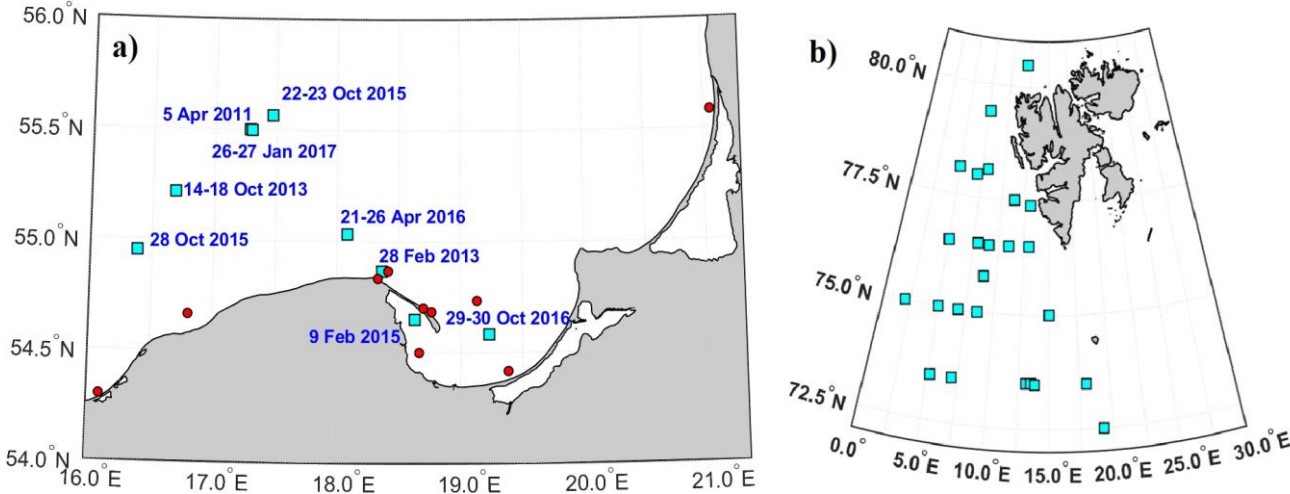

**Figure 1. The flux measurement stations in the a) Baltic Proper area (left panel), and in the b) North Atlantic Ocean. Stations marked with a cyan colour indicate a long fetch and sea air mass conditions which were used in the analysis. Points marked with a red colour indicate a short fetch and land air mass advection which were excluded from the analysis (these factors were obtained based on the Hysplit backward trajectories – see the Appendix A).**

## 3.2 Instrumentation

For aerosol measurements the optical particle counter CSASP-100-HV (CSASP) of the particle measurement system was used. The device counts aerosol particles in the diameter range $0.5 < D_p < 47$ μm. In this study, 36 particle size bins were used for analysis. The technical specification of this instrument is given in the Appendix B.

Meteorological parameters such as wind speed and humidity were measured with a Vaisala meteorological probe at 10 meters above the sea surface (WXT530). Humidity measurements were also with regular intervals verified using an Assman 255 psychrometer.

## 3.3 Gradient flux determination

To apply the gradient method, CSASP was placed on a special lift in the starboard side of the ship. The lift allows to change the height of the measurements between 5 levels: 8 m, 11 m, 14 m, 17 m, 20 m. A single measurement on a single level lasts at least 2 minutes. After this time, the device was moved to the next level. Each flux was determined based on 30 minutes of 260 such measurements, based on the vertical aerosol gradient, friction velocity, and stability. The application of the method is presented in the section C of the Appendix and detailed description of measurement methodology is given by Petelski (2003).





The net fluxes derived from the gradient method are affected by upward (emission), and downward (deposition) fluxes. This study focus on the SSA emission, so all presented fluxes are emission fluxes and it is labelled as $F_N$ (flux of aerosol volume is marked as $F_V$). SSA emission fluxes were obtained based on subtraction of modelled deposition fluxes from the net flux; in

this respect, we used the Schack et al. (1985) model for deposition flux as it has been tested previously for both Arctic waters (Nilsson et al., 2001) and the Baltic Sea (Nilsson et al., 2021). Due to the size range of the CSASP (>0.5 µm $D_p$), the deposition flux from Schack et al. (1985) was dominated by impaction and sedimentation. All fluxes were reduced to 80 % equilibrium humidity using Fitzgerald (1975) approach updated with the recent sea spray growth factor given by Zieger et al. (2017). In this paper all aerosol and flux spectra are presented in relation to reduced diameters, which was marked later as $D_{p@80\%}$ -

diameter at 80 % of humidity.

### 3.4 External data sources

#### 3.4.1 Reanalysis data

In this work global atmospheric reanalysis (Hersbach et al., 2018) was used available from the Copernicus platform. The parameters used are available with a 1 hour temporal resolution and a spatial resolution of 0.5×0.5 degree. They included:

wind speed ($U_{10}$), significant wave height ($H_S$), spectral peak period ($t_p$), sea surface temperature ($T_w$), and air temperature ($T_a$). For the Baltic region, wave data available as a part of the E.U. Copernicus Marine Service Information (Baltic Sea Wave Hindcast) were used. Wave data such as significant wave height $H_s$ and spectral peak period $t_p$ calculated by the WAM model cycle 4.6.2 are available with 1×1' spatial resolution and with 1 hour temporal resolution.

In order to obtain the same temporal resolution as fluxes (0.5 h) we interpolated all data for each a half hour by averaging two

known data points for each full hour.

Chlorophyll-$a$ (Chl-a) - as an indicator of marine biological activity - for the Atlantic Ocean were obtained from satellite data available in Global Ocean Colour (Copernicus-GlobColour, Garnesson et al., 1997).

Chl-a data in the Baltic region were obtained from the SatBałtyk system (Satelite Monitoring of the Marine Environment in the Baltic Sea, Woźniak et al., 2011a, Woźniak et al, 2011b) which is run by the Institute of Oceanology Polish Academy of

Sciences. The Chl-a data is the product of satellite data (MODIS satellite) compiled with EccoSat ecohydrodynamic model (Kowalewski, 2015). The approach of data compilation is presented by Konik et al., (2019).

#### 3.4.2 Back trajectories

We established air mass backward trajectories, fetch, and time over the sea during measurements. To determine these, we used the Hysplit model (Stein et al., 2015) fed by NCEP GDAS (The National Centre for Environmental Prediction, Data

Assimilation System) 0.5×0.5 degree meteorological data; the back trajectories were propagated for 24 h reaching a final altitude of 10 m a.s.l. Hysplit outputs also embedded the mixing layer depth. All measurement time series with date, time, duration, are presented in the Appendix A (Table A1 and A2).



### 3.4.4 Secondary parameters

The meteorological and oceanographic parameters were compared with the SSA fluxes. Based on primary parameters ($U_{10}$,

$H_S$, $t_p$, $T_w$, $T_a$), we also calculated secondary parameters such as: wind acceleration $a_U$, (eq. 2), wave age (eq. 3), wave Reynolds number (eq. 5), bulk atmospheric stability (also known as temperature difference between seawater and air): $T_d = T_w - T_a$. In order to calculate water viscosity in $ReH_w$ determination we used seawater properties library routines given by Nayar et al., (2016) and Mostafa et al., (2017).

### 3.5 Error propagation

Measurements uncertainty values were calculated according to general rules of error propagation (i.e., Taylor, 2012). The absolute uncertainty of the particle counter can be obtained from Poisson's distribution properties, which is the standard deviation $\sigma_P = \sqrt{\mu}$, where $\mu$ is mean counting value. The relative uncertainty can be defined as $\mu^{1/2}/\mu$ so it is inversely proportional to the mean particle counts (in our case the single measurement was 10 s). For counts in the order of magnitude $n \sim 10^3$, m$^{-2}$ the relative uncertainty is $\Delta n \sim 10$ % for the particles range 0.5–2.5 µm, increasing to 31 % for the particle size

range 2.5-7 µm and reaching a value up 90 % for particles larger than 7 µm. By increasing the averaging time up to 30 min, we can decrease the uncertainty, up to $\Delta n \sim 10$ % which is in the same range as for other OPCs available on the market (i.e., TSI 3340 LAS, Grimm OPC 1.109).

The uncertainty of wind speed measured by acoustic anemometer according to the manual is below <1 %. The uncertainty of the data taken from the WAM model is estimated as follows: $\Delta U_{10} \approx 10$ %, $\Delta H_s \approx 2$–5 %, after Janssen et al. (2007), $\Delta\omega_p \approx 15$

% after Abdalla et al. (2010).

The uncertainty of the fitting $N_*$ parameter can be obtained from absolute uncertainty of the least squares method:

$$\sigma_n = \sqrt{\frac{1}{l-2}\sum_{i=1}^{l}\left(n_i - C - N_* ln(z)\right)^2}, \tag{6}$$

$$\sigma_{N_*} = \sigma_n \sqrt{\frac{l}{l\sum ln(z)^2 - (\sum ln(z))^2}}. \tag{7}$$

Where $l=5$ (number of measurement levels), $ln(z)$ is the natural logarithm of measurement height, $n_i$ is a single measurement

on each level. For different values of the correlation coefficient $r$ of the fitting, relative uncertainty (defined as $\sigma_{N_*} N_*^{-1}$) were checked respectively. For $r$ values in the range $1.0 < r < 0.5$ uncertainty was lower than 3 %. For worst fitting ($r < 0.5$) the relative uncertainty increases drastically (for $r = 0.2$ even 30 %). Lower values of $r$ imply that aerosol concentration gradient did not exist, so it is impossible to consider the aerosol gradient flux. As such, data with correlations lower than 0.4 were excluded from further consideration.




## 4 Results and discussion

### 4.1 Fluxes overview

A histogram of total fluxes $F_N$ is presented in the Appendix (Fig. A1) from which it is clear that positive (emission) fluxes dominated in both Baltic and Atlantic data. The peak for the Baltic Sea fluxes at 5 % of total frequency occurrence have a median value $M_F$=3.4·10$^4$ m$^{-2}$ s$^{-1}$ (mean value $\mu_F$=8.9·10$^3$ m$^{-2}$ s$^{-1}$) and standard deviation $\sigma_F$=6.8·10$^5$ m$^{-2}$ s$^{-1}$. For the Atlantic data set, the peak of occurrence is at 14 % of cases with a median value $M_F$=4.0·10$^4$ m$^{-2}$ s$^{-1}$ (mean value $\mu_F$=1.0·10$^4$ m$^{-2}$ s$^{-1}$) and $\sigma_F$=2.7·10$^5$ m$^{-2}$ s$^{-1}$. The significantly higher fluxes measured in the Atlantic data set is explained by the fact that during ocean measurements we deal with high fetch and lack of air mass advection from land. In the Baltic Sea cases, measurements with land advection (according to the 24 h backward trajectories) were excluded from further analysis.

To avoid influence of air mass advection from land, the measurements were divided into two groups according to surface conditions according to 24 h backward trajectories (see below chapter 3.5.2): 1) entirely marine the last 24 h, and 2) some marine, with land influence during the last 24 h. For marine conditions, the estimated fetch varied between 110 km and 350 km. The times over the sea for marine conditions varied from 6 to 20 h.

### 4.2 Overall data trends

In order to present the large data trends in a compact way, we use the following exponential fits to describe the relationship between total aerosol fluxes and wind speed, wave age, wave height and wave Reynolds number:

$$P(x) = \exp(ax^b), \tag{8}$$

The relationship between wind acceleration and aerosol number flux could be described with the following exponential relationship:

$$G(x) = \exp(ax + b), \tag{9}$$

The relationship between temperatures and aerosol number flux could be described with the following linear relationship:

$$L(x) = ax + b, \tag{10}$$

where $x$ stands for a chosen parameter. The fitting results with the functional parameters $a$ and $b$ are presented in the Table 1 and Table 2. The first two functions $P(x)$, and $G(x)$ were used for the total fluxes. The third group of functions $L(x)$ were fitted to fluxes which were normalized by a wind speed relation:

$$F_{sn} = F_N/P_B^\beta(U_{10}). \tag{11}$$

the resulting coefficients of this parameterisation are presented in the Table 3.

The Tables 1-3 also contain the squared correlation coefficient. We can note a strong correlation between the total aerosol flux and $U_{10}$ at low Chl-a, total aerosol flux and $ReH_w$.



**Table 1. Parameterization results between several ambient factors and the total aerosol fluxes from the Baltic Sea: wind speed $U_{10}$, wind acceleration $a_U$, wave age $c_p/U_{10}$, Wave Reynolds Number $ReH_w$ and Significant Wave Height $H_s$. The parameterisation coefficients $a$ and $b$ based on equations 8, 9 and 10, $r$ represents fitting correlation. Additional subscripts indicate as follows: 'B' - the fit to the Baltic Sea data, '$\alpha$' - fit to all range of Chlorophyll-$a$ concentrations, '$\beta$' – fit to data in range below 3.5 mg m$^{-3}$ of Chl-a, and ' $v$ ' – fit to volume flux data.**

| Source function | | $a$ | $b$ | $r$ |
|---|---|---|---|---|
| $P_B^{\alpha}$ $(U_{10})$ | All data | 6.95 | 0.23 | 0.57 |
| $P_B^{\beta}$ $(U_{10})$ | Chl-a<3.5 mg m$^{-3}$ | 5.45 | 0.35 | 0.87 |
| $P_{vB}^{\alpha}(U_{10})$ | All data | 9.64 | 0.20 | 0.68 |
| $P_{vB}^{\beta}(U_{10})$ | Chl-a<3.5 mg m$^{-3}$ | 9.11 | 0.23 | 0.74 |
| $G_B(a_U)$ | Chl-a<3.5 mg m$^{-3}$ | 10409 | 11.75 | 0.68 |
| $P_B(c_P/U_{10})$ | | 11.21 | -0.24 | 0.58 |
| $P_B(ReH_w)$ | | 2.60 | 0.12 | 0.77 |
| $P_B(H_s)$ | | 11.58 | 0.15 | 0.63 |

**Table 2. Parameterization results between factors and total aerosol fluxes from the Atlantic Ocean. The analysed parameters are the same as in Table 1. Subscript 'A' indicates fit to the Atlantic Ocean data.**

| Function | $a$ | $b$ | $r$ |
|---|---|---|---|
| $P_A(U_{10})$ | 8.06 | 0.16 | 0.43 |
| $G_A(a_U)$ | 4187 | 11.12 | 0.37 |
| $P_A(c_p/U_{10})$ | 11.37 | -0.07 | 0.42 |
| $P_A(ReH_w)$ | 4.75 | 0.07 | 0.43 |
| $P_A(H_s)$ | 10.83 | 0.11 | 0.37 |

**Table 3. The fitting coefficients of functions fitted to all wind speed-normalized aerosol fluxes according to the equation 11.**

| Functions | The Baltic Sea | | | The Atlantic Ocean | | |
|---|---|---|---|---|---|---|
| Baltic, Atlantic | $a$ | $b$ | $r$ | $a$ | $b$ | $r$ |
| $L_B(T_a)$, $L_A(T_a)$ | -0.13 | 2.14 | 0.33 | 0.04 | 1.11 | -0.06 |
| $L_B(T_w)$, $L_A(T_w)$ | -0.20 | 2.70 | 0.49 | -0.10 | 1.96 | 0.12 |
| $L_B(T_d)$, $L_A(T_d)$ | 0.23 | 1.55 | 0.30 | -0.25 | 1.69 | 0.23 |




### 4.1.1 The impact of marine biological activity on sea spray fluxes

Based on the findings by Nilsson et al. (2021), who reported a weakened wind dependency when organics were present, we could expect influences of the biological activity in the seawater on sea spray formation. Some of our measurements in the Baltic region were also carried out during early summer and autumn, the typical periods of algae bloom in the Baltic Sea. In contrast, our Atlantic measurements were carried out entirely during the summer season. For this reason, we chose to employ Chl-a as a proxy for marine biological activity. From here on, we use superscript $\alpha$ to denote the entire data set, and superscript $\beta$ for Chl-a < 3.5 mg m$^{-3}$.

In the Baltic Sea the Chl-a varied between values of 0.23 mg m$^{-3}$ and 4.28 mg m$^{-3}$. In the Atlantic Ocean measurements minimum Chl-a concentration was 0.13 mg m$^{-3}$ and maximum 2.20 mg m$^{-3}$. These numbers are within the range of typical seasonal mean values (Stoń-Egiert and Ostrowska, 2022). The upper range compares also well with the Baltic EC flux data by Zinke et al. (2024), with averages of 3.9 and 5.2 mg m$^{-3}$, for May and August field campaigns, respectively.

In the Fig. 2 we present interrelations between total SSA fluxes, wind speed and water Chl-a concentration in the Baltic Sea. To indicate the impact of Chl-a, we fitted four curves to our data set assigned respectively as $P_B^\alpha(U_{10})$, $P_B^\beta(U_{10})$, $P_{vB}^\alpha(U_{10})$, and $P_{vB}^\beta(U_{10})$: red dashed lines for the entire data set (upper index α), and blue dotted lines for low Chl-a (below 3.5 mg m$^{-3}$ (upper index β). The first two functions were fitted to number flux and second two were fitted to volume flux (as indicated by the additional subscript 'v'). The curves coefficients are specified in the Table 1.

All measurements at higher Chl-a concentration were made in the open sea region (5.05.2011, 18.10.2013, 23.10.2015, 22.04.2016 – details with mean values are presented in the Table A1 of the Appendix A). All these high Chl-a measurements were made in the open sea region without any fresh water inflow from rivers that might have affected the measurements. This is also reflected in the constant salinity values across all stations.

In the wind range 8.5-10.5 m s$^{-1}$ we observed higher aerosol emissions at Chl-a>3.5 mg m$^{-3}$ compared to lower Chl-a concentrations. While, at higher wind speeds (10.5 m s$^{-1}$ - 14.5 m s$^{-1}$) with high Chl-a, we observed almost an order of magnitude lower aerosol fluxes than in the case of low Chl-a concentrations. This effect is less pronounced if we investigate the SSA volume fluxes (Figure 3b), which suggest that smaller particles were mostly associated with higher emission. We did not observe such an effect on the fluxes measured in the Atlantic Ocean.

To statistically prove the difference between the fits to the low and high Chl-a regimes ($P_B^\alpha(U_{10})$ with $P_B^\beta(U_{10})$, and $P_{vB}^\alpha(U_{10})$ with $P_{vB}^\beta(U_{10})$) we applied several statistical tests (all tests' results are presented at the 5 % significance level). Since the variances of analysed data sets are not equal, we have to use the so-called unequal variances t-test (Welch's t-test which is the generalization of the classical t-test). A detailed summary of the test results is provided in table D1 of the Appendix D .

In terms of number flux, the low and high Chl-a regimes were statistically different ($p$=0.0094), while in terms of volume flux the difference was not significant ($p$=0.13). This finding implies that emission of smaller sub-micrometre particles (which contribute into number concentration) was more affected by Chl-a than bigger super micrometre particles (which contribute into aerosol volume).





We also employed the right tailed Welch's test in order to investigate in detail the difference between curves $P_B^\alpha(U_{10})$ and

$P_B^\beta(U_{10})$. The fits have been segregated into low wind speed range (below 10 m s$^{-1}$) and higher wind speed range (above 10 m s$^{-1}$). For low wind speeds case the right tailed Welch's test proved that that there was no significant difference between both fits ($p$=0.32). However, at high wind speeds they were significantly different (p=5·10$^{-4}$ ).

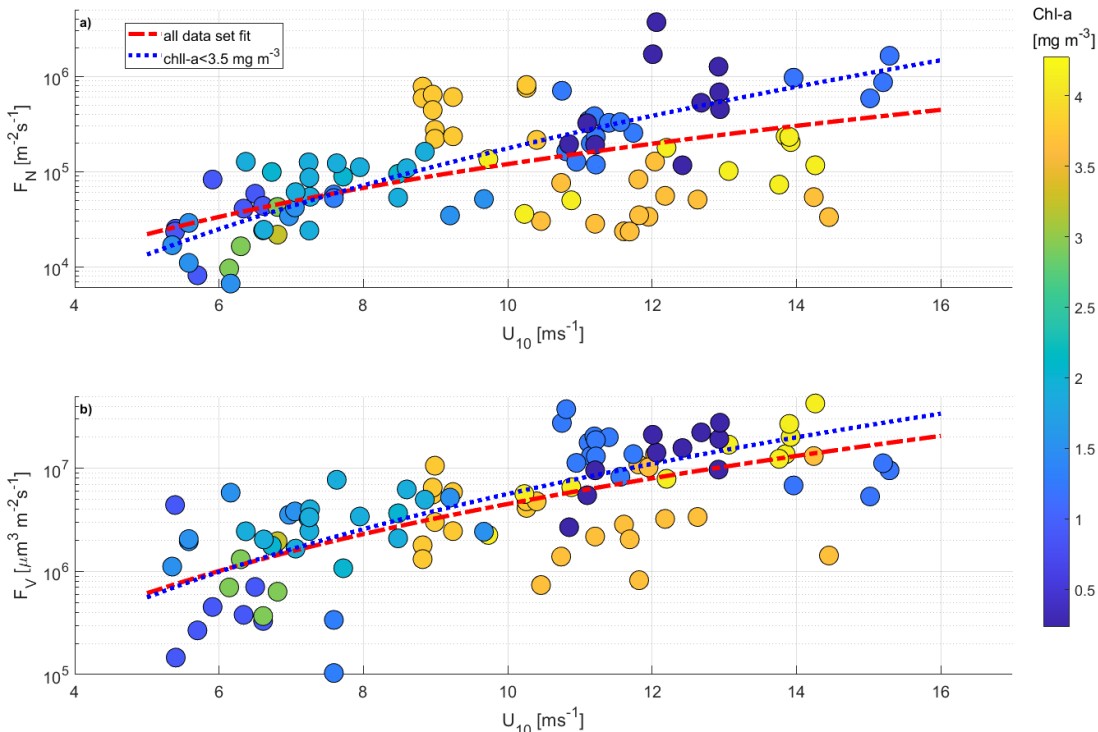

**Figure 2. The influence of 10 m wind speed and chlorophyll-*a* concentration in water on sea spray fluxes over the Baltic Sea.2a)**
**Aerosol number flux. 2b) Aerosol volume flux.**

The above result showed that number concentration flux from waters with higher Chl-a was decreased in higher wind speed regimes in comparison with low Chl-a concentrations. This finding suggests that biological activity indicated by Chl-a may have a suppressing effect on aerosol number flux. The cause of flux decrease for higher organic activity may be caused by changes in surface tension which affect the bubbles properties such as bubble film thickness and lifetime (Sellegri et al., (2021),

Barthelmeß and Engel (2022).

Simultaneously, we did not observe a dampening effect for the volume flux which is mostly affected by bigger particles. This result is related with previous studies which reported that submicron particles emitted from the sea surface are mostly enriched with organic matter while super micron particles mainly consist of inorganic sea salt (Keene et al., (2007), Yoon et al., 2007, Facchini et al., (2008), Quinn et al., (2014)). For this reason, we should not expect a significant influence on the gradient





aerosol volume flux from organic sea spray in the super micrometre range of the CSASP-100-HV used in this study, and only limited influence in the sub micrometre range, since it is limited to >0.5 μm $D_p$.

Nonetheless, Nilsson et al. (2021, figure 9) reported that while the EC flux of sea salt particles had a log-linear fit between $\log_{10}(F_n)$ and $U_{10}$ with a slope of 0.11 and correlation of $r=0.74$, while the EC flux with organic sea spray included only gave a slope of 0.02 and correlation $r=0.23$. Hence, the presence of organic compounds in the sea spray aerosol decreases  SSA

emissions, and at the same time increase the scatter of the data points, lowering the correlation. This is in good agreement with Fig. 2, especially the gradient aerosol number flux in Fig. 2a, where the slope decrease significantly when high Chl-a data points are included, and the scatter between data points increase.

### 4.1.2 Parameters correlations with sea spray emission

In Figure 3 we present Spearman's correlations between measured total fluxes (Fig. 4a) and volume fluxes (Fig. 4b) with

analysed parameters, and statistical significances expressed as p-values.

As described above, we have split the Baltic Sea measurements into two categories: data with low Chl-a concentrations (Chl-a < 3.5 mg m$^{-3}$, see Fig. 4a) and all data. We will treat the Atlantic Ocean data as a 3$^{rd}$ category, where Chl-a was always lower than in the Baltic Sea data (Atlantic Chl-a = 0.60 ± 0.58 mg m$^{-3}$). We observed the highest, statistically significant (p-value<0.05), positive correlations ($r_s$>0.6) between the gradient aerosol fluxes and $U_{10}$ and wave state parameters at low Baltic

Sea Chl-a conditions

The effect of biological activity in the Baltic Sea region reduces correlation of wind and wave parameters with aerosol fluxes, which is in agreement with Nilsson et al. (2021). The influence of biological activity hence weakens the effect of increasing $U_{10}$. This is why, for the remaining analysis we decided to present the effect of $U_{10}$ and wave state by analysing only aerosol gradient fluxes with low Chl-a.

In middle and high latitude meteorological data the correlation between wind speed and temperature is usually high, because cold periods usually are associated with higher wind speed and vice versa, on both a seasonal and a synoptic time scale. Likewise, sea surface temperature and the temperature of the atmospheric surface layer is usually highly correlated. In order to eliminate the influence of wind speed on our data which may interfere with temperature effects we calculate correlations between normalized aerosol fluxes and temperature parameters ($T_a$, $T_w$, and $T_d$). We implemented a normalization by dividing

measured fluxes by obtained wind speed related function (details in the section 4.3.3).

For the effect of temperature, we observed a more complex picture. We found that $T_a$ and $T_w$  have a negative statistically significant correlation ($r_s$<-0.5, p-val<$10^{-11}$) with all Chl-a cases of the Baltic aerosol number fluxes (Fig. 3a). The remaining calculated correlations (cases of low Baltic Sea Chl-a, and the Atlantic Ocean which also had low Chl-a conditions) were insignificant (and also negative apart from $T_d$ in the Atlantic Ocean cases). We obtained the opposite results when analysing

the aerosol volume flux. In this case, $T_a$ and $T_w$ correlated positively with Baltic Sea gradient aerosol volume fluxes limited to low Chl-a conditions. The correlations were not high ($r_s$~0.4), but in both cases significant (p-val<0.05). The cases of high





Chl-a correlations were insignificant for $T_a$ and $T_w$. We also obtained statistically significant negative correlations between aerosol gradient volume fluxes and $T_d$ in all three data sets.



**Figure 3. Spearman correlations calculated between measured aerosol fluxes and investigated parameters (wind speed $U_{10}$, wave Reynolds Number $ReH_w$, wind acceleration $a_U$, wave age $c_p/U_{10}$, wave height $H_s$, air temperature $T_a$, water temperature $T_w$. In the panel a we presented number fluxes correlations and in the panel b volume fluxes correlations. Circles represents correlations in the Baltic Sea in all Chl-a conditions, diamonds represent correlations in the Baltic Sea in low Chl-a conditions, stars represent Atlantic Ocean's correlations. Statistical significances are presented as p-values with colours. Blue colour represent low p-value so the correlation is statistical significant, green colour is boundary of statistical significance level. The yellow colour indicates high p-values so the correlation is insignificant.**



### 4.1.3 Dependence of sea spray fluxes on horizontal wind speed

An exponential increase in the total SSA number flux was observed with increasing horizontal wind speeds and wind acceleration for both the Baltic Sea and Atlantic Ocean measurements (see Fig. 4).

SSA fluxes were categorized based on wind speed classes, each having a bin width of 2 m s$^{-1}$. Aerosol gradient fluxes measurements were carried out over the Baltic Sea between 2.1 m s$^{-1}$ up to 15.8 m s$^{-1}$, in the Atlantic Ocean between 2.1 m s$^{-1}$ to 11.9 m s$^{-1}$.


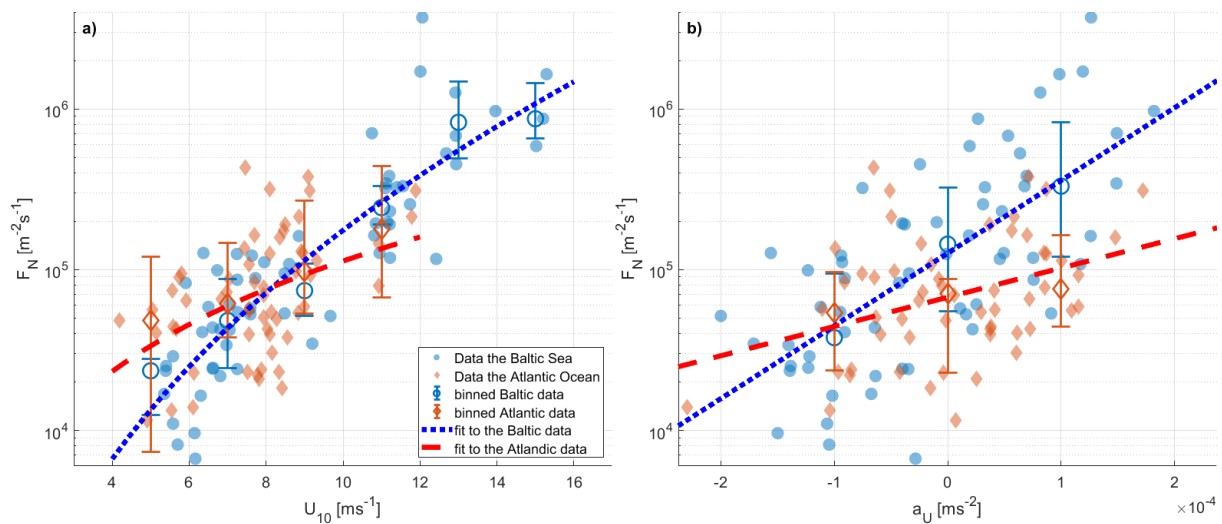

**Figure 4. Total aerosol fluxes, versus wind speed (panel a) and wind acceleration (panel b). Error bars represent first and third quantile of each data bin (25-75 %) with a median value (50 %). Fitting coefficients are presented in Table 1.**

For each wind speed class we present median and percentile 25-75 % with empty circles for the Baltic Sea data and empty
diamonds for the Atlantic Ocean data and error bars. We used the same convention for all total flux comparisons with other parameters.

Size resolved aerosol gradient fluxes binned according to these wind speed classes are presented in Figure 5. In the Baltic Sea we can see a systematic increased emissions of particles from 0.25 μm < $D_p$ < 2 μm. For aerosols with $D_p$ > 2 μm, the pattern is different: in the highest wind speed class (15 m s$^{-1}$ ) we observed a small decrease in the emission fluxes compared to 11 m
s$^{-1}$.

In Figure 5b we present aerosol gradient flux size distributions in wind speed classes present in the Atlantic Ocean data (there were no $U_{10}$ in the 15 m s$^{-1}$ range in the Atlantic Ocean data set). In this case the increase in fluxes with the wind speed was smaller, which is also visible if we compare the slopes of the fitted curves in Figure 4a). We can also observe that in lower wind speed classes the measured Atlantic Ocean emission aerosol gradient fluxes were higher than over the Baltic Sea, by
factors of 3.0 – for 5 m s$^{-1}$ and 1.7 for 7 m s$^{-1}$. The highest $U_{10}$ range present in both data sets, and both Figures 5a and 5b (11 m s$^{-1}$) are relatively similar, so the smaller increase in sea spray emissions over the Atlantic Ocean with increasing $U_{10}$ is due to higher fluxes at the lowest wind speeds.





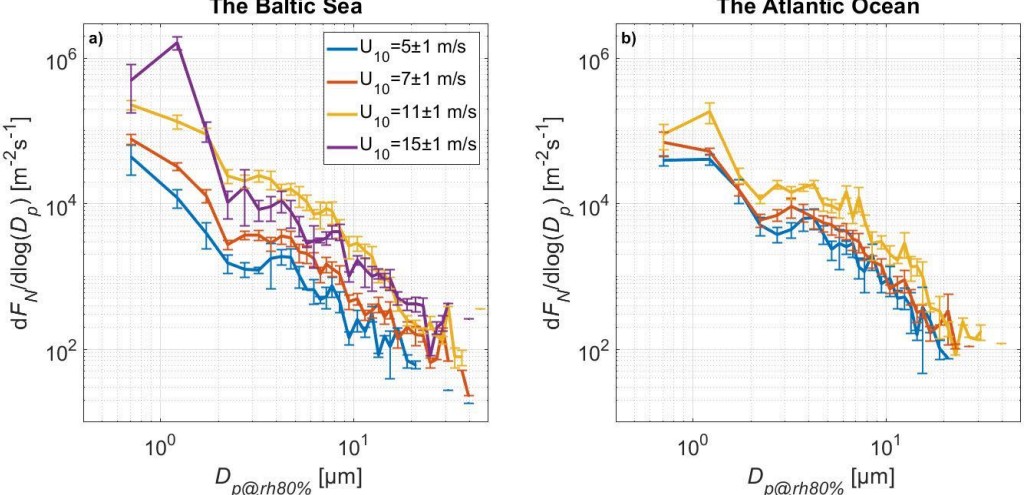

**Figure 5. Measured mean sea spray aerosol number gradient flux spectra reduced to 80 % relative humidity in relation to wind speed binned in wind speed classes ($U_{10}$). Error bars indicate one standard error. 5a) Over the Baltic Sea. 5b) Over the Atlantic Ocean. The highest wind speed range ($U_{10} = 15$ m s$^{-1}$) is absent in the Atlantic Ocean data set (5b). No $U_{10} < 4$ m s$^{-1}$ is included since we do not expect breaking waves in this wind speed range.**

### 4.1.3 Dependence of sea spray fluxes on horizontal wind acceleration

The wind acceleration varied from minimum values $-2.0 \cdot 10^{-4}$ m s$^{-2}$ (over the Baltic Sea) and $-2.3 \cdot 10^{-4}$ m s$^{-2}$ (over the Atlantic Ocean) up to maximum values $1.8 \cdot 10^{-4}$ m s$^{-2}$ and $1.7 \cdot 10^{-4}$ m s$^{-2}$. In this case we assumed only three classes – decreasing wind speed $a_u < -2.5 \cdot 10^{-5}$ m s$^{-2}$, increasing wind speed $a_u > 2.5 \cdot 10^{-5}$ ms$^{-2}$ and near constant wind speed $-2.5 \cdot 10^{-5} < a_u < 2.5 \cdot 10^{-5}$ m s$^{-2}$. Over the Baltic Sea, the measured aerosol gradient flux size distributions for increasing $U_{10}$ conditions are higher for decreasing wind conditions in the sizes up to the diameters of 16 µm. For larger particles, the difference between these two classes is much smaller and less significant (overlapping each other's error range). We did not observe any difference between these two classes over the Atlantic Ocean. The reason for that may be the lack of $U_{10} > 12$ m s$^{-1}$ (and hence less wind speed accelerations), which don't let us observe enough variability.



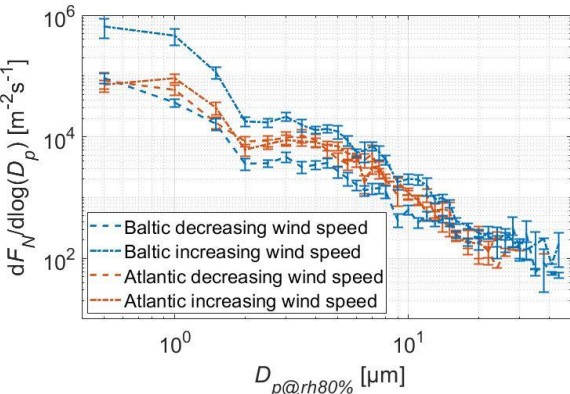

**Figure 6. Aerosol number sea spray gradient flux size distribution in relation to wind acceleration parameter $a_u$. The size spectra**
**presents mean values of over the Baltic Sea and Atlantic Ocean for each of two cases: increasing $U_{10}$ ($a_u > 2.5 \cdot 10^{-5}$ m s$^{-2}$) and decreasing**
**wind speed $U_{10}$ ($a_u < 2.5 \cdot 10^{-5}$ m s$^{-2}$).**

This is opposite to the total aerosol fluxes (0.11-6 µm $D_p$) reported by Yang et al. (2019), which is larger for decreasing $U_{10}$,

than for increasing $U_{10}$. In the current study this could be explained by larger $H_s$ during decreasing wind speeds. This is also

opposite to the wind history effect on $W$ presented by Stramska and Petelski (2003) and Callaghan et al. (2008), which showed

that $W$ was larger for the same wind speed when the wind speed was decreasing than when it was increasing, but only for $U_{10}$

> 10 m s$^{-1}$. This data set was from a ship cruise in the north-east Atlantic Ocean, so it should be more comparable to our

Atlantic data set. We are not aware of any whitecap wind history study in the Baltic Sea.

### 4.1.4 Wave state parameters

For the Baltic Sea cruises it was possible to gather a wide range of wave heights ranging from 0.1 m up to 3.7 m. The

corresponding peak period ranged from 1.5 to 9.3 s. In the Atlantic Ocean region, the $H_s$ value varied in the range 0.7 to 2.2

m. Peak period values were in the range of 4.7 to 8.2 s.

With increasing wave age, we observe an exponential decrease in total aerosol production (see Figure 7a). In the Baltic area

our SSA gradient flux observation covered wave ages in the range 0.46 < $c_p/U_{10}$ < 1.87. In the Atlantic Ocean region, there

were no younger waves, therefore, observations covered wave ages in range 0.81 < $c_p/U_{10}$ < 8.23. Measurement points were

combined in wave age classes with a bin width of 0.5: for the Baltic Sea data - three classes, for the Atlantic data - six wave

age classes.

We can see a constant decrease of the aerosol flux with the wave age in both data sets. We observed much more rapid decrease

of the total gradient aerosol flux over the Baltic Sea than over the Atlantic Ocean, and $c_p/U_{10}$ span over a wider range, with

higher values.



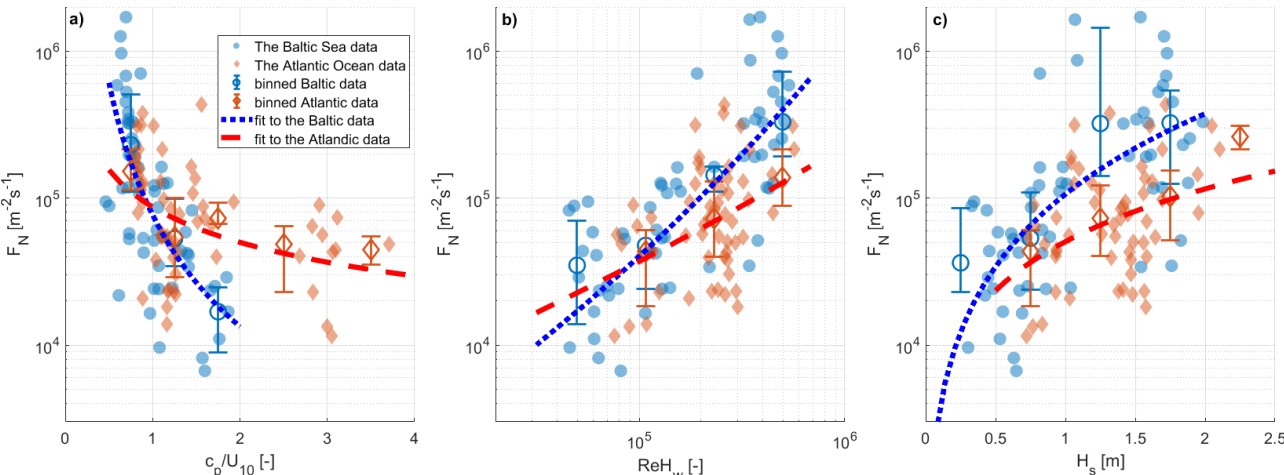


**Figure 7. Total aerosol number gradient fluxes in dependence of wave age (a), wave Reynolds number $ReH_w$ (b) and wave height $H_s$ (c). Error bars represent first and third quantile of each data bin (25-75 %) with a median. Fitting coefficients are presented in the Table 1.**

Both data sets were binned into four bins of $ReH_w$ (Fig. 7b). We observed a linear increase of the total aerosol gradient number

flux with increasing $ReH_w$ both over the Baltic Sea and Atlantic Ocean. This increase was steeper over the Baltic Sea. In the highest bin of the Reynold Number (mean $ReH_w = 5·10^6$) we observed higher aerosol flux in the Baltic Sea than over the Atlantic Ocean. Yang et al. (2019) presented total aerosol EC fluxes (0.11-6 μm $D_p$) as a function of $H_s$, and for their open sea data they derived a linear relationship, for $H_s$<1 m and EC fluxes <$10^4$ m$^{-2}$s$^{-1}$. Zinke et al, (2024) presented a total aerosol EC flux (0.25-2.5 μm wet diameters) as a power law function of $H_s$, where the slope was 0.05 and the power 2.2. The aerosol

EC fluxes were <$3·10^5$ m$^{-2}$s$^{-1}$, and $H_s$ < 2 m. Yang et al. (2019) noted that $ReH_w$ fail to reconcile their EC aerosol fluxes with those of Norris et al. (2013), and we can now conclude that neither reconcile this data set, or that of Zinke et al. (2024) with these sea spray fluxes.

Figure 7c shows the total aerosol gradient fluxes binned according to wave height $H_s$ with a bin width of 0.5 m. We observed a SSA flux increase with increasing $H_s$ which was also reported by Yang et al., (2019) and Zinke et al., (2024). They have

aerosol EC fluxes of higher magnitudes than our gradient aerosol fluxes because their measurements included smaller aerosol particles. Zinke et al. (2024) reports a linear fit between aerosol EC emission fluxes with a slope of $1.68·10^{-7}$. Yang et al. (2019) presented and exponential fit, with unknown slope. The curves fitted to our data were exponential power laws, where the slope a was 2.60 and the power 0.12.

### 4.1.5 Temperatures and Atmospheric Stability

We further investigated the impact of air and seawater temperatures on the aerosol emission flux. The air temperature oscilated in the Baltic Sea from -3.2 °C up to 13.3 °C and in the Atlantic Ocean from 1.2 °C up to 8.4 °C. The water temperature in the Baltic Sea was between 1.6 °C to 13.1 °C, and in the Atlantic Ocean between 2.3 °C to 8.4 °C. We also used the temperature difference between sea surface temperature and air temperature at 2 m above sea level ($T_d=T_w-T_a$) as an indicator of



atmospheric stability. These values varied from -2.6 ℃ up to 4.8 ℃ in the Baltic Sea, and -0.5 ℃ up to 4.0 ℃ in the Atlantic
Ocean region.

Due to a correlation between temperature and wind speed, a direct correlation between aerosol gradient fluxes and temperature
could be biased. The reason is the strong high correlation between $T_a$ and $U_{10}$ on both seasonal and synoptic time scales (higher
air temperatures correlates with lower wind speeds, and vice versa). That is why, in order to exclude the effect of $U_{10}$ we
normalized fluxes by our source function expression $P_B^\beta$ $(U_{10})$. The normalized fluxes, in relation to $T_a$, $T_w$ and $T_d$, are presented
in the Fig. 8.

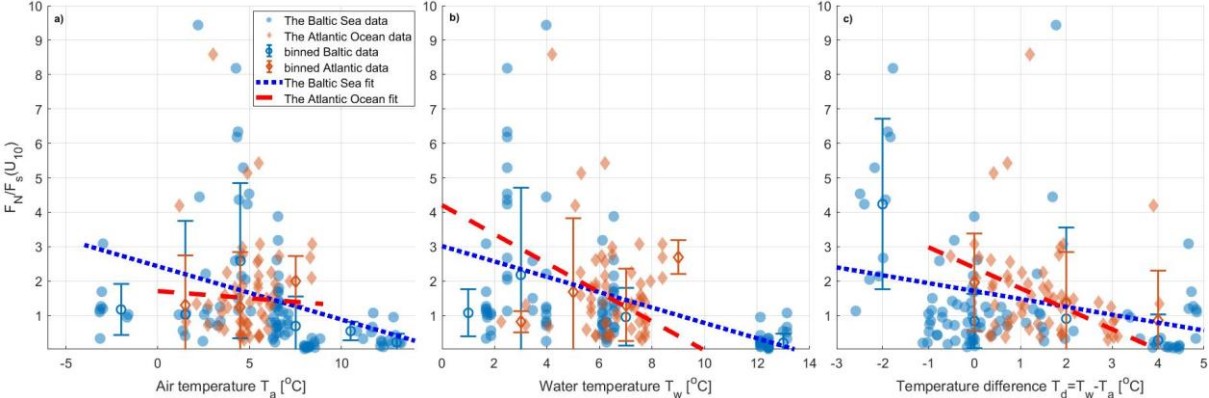

**Figure 8. Total sea spray aerosol gradient fluxes normalized by the wind speed dependent source function $P_B^\beta$ $(U_{10})$ (Eqs. 8 and 11)
as function of air temperature $T_a$, water temperature $T_w$ and temperatures difference $T_d$. We fitted the functions according to the
equation 10. The fitting coefficients are presented in the Table 3.**

As we can see in the Figure 3 and Table 3, the Spearman's correlations of wind normalized fluxes with $T_a$ and $T_d$ are lacking
significant aerosol volume flux correlations. In case of aerosol gradient number fluxes we observed negative correlations
(statistically significant only in case of all Baltic Sea Chl-a values) and only for $T_w$ ($r^2$=0.24), using a linear fit with a negative
slope of -0.20, see Table 3. In case of volume fluxes we observed positive correlations (statistically significant only for low
Chl-a case, see Fig. 3).

The decrease in sea spray emission gradient fluxes with increasing $T_w$ are consistent with studies which report suppression of
SSA emission with increasing $T_w$. (see references in the Introduction). Compare also with Figure 3. It is worth noting that our
Baltic Sea data set, which spans from $T_w$=1.6℃ to 13.1℃ covers exactly the $T_w$ range (from -2 ℃ to about 12-15 ℃) in which
Salter et al. (2014) and many others have observed the largest decrease in sub micrometre sea spray production. The narrower
$T_w$ range in the Atlantic data set (2.3 ℃ to 8.4 ℃) may not be enough to distinguish this trend. The fact that we observe it only
for the total aerosol number gradient flux is logical, since this should be dominated by the sub micrometre aerosol particles,
while we don't see a clear trend in the total aerosol volume gradient fluxes, since these are likely dominated by super
micrometre particles.

As discussed in the Introduction, for the super micrometre sea spray, some studies show an increase in sea spray emissions
with increasing $T_w$ (Liu et al., 2020). So, in this case our results also support this finding. Based on that we speculate that we





observe two different phenomena (just like Bowyer et al. 1990). An increase in $T_w$ suppresses the emission of submicron particles but increases the emission of super micrometre mode. This leads to a decrease in SSA number but an increase in SSA volume with increasing $T_w$.

The higher aerosol gradients observed under stable atmospheric conditions are likely caused by increased stratification and suppression of turbulent mixing and dispersion. In case of stable atmosphere turbulent eddies were smaller and less energetic,
so particles reached lower heights, which caused that we observed higher gradient flux. As we can see, our data source fittings have low correlations. Large data scatter suggests that many effects interact in these observations. Further research is needed to disentangle the different factors impacting SSA fluxes.

**4.3 Comparison with relevant source functions**

We decided to choose several different source function for comparison with our data sets. In order to compare wind speed
dependence, we used three functions determined by in-situ measurements: Petelski and Piskozub (2006) with Andreas (2007) improvement - $f_{pet}$, Smith et al., (1993) - $f_{sm}$, and Zinke et al. (2024) $f_{zin}$. The $f_{zin}$ is the only one specifically derived for the brackish sea water of the central Baltic Sea, and therefore relevant for part of our data set. We also used two functions determined in tank experiments: Mårtensson et al., (2003) - $f_{mar}$, and Salter et al., (2015) - $f_{sal}$, because they are currently the only ones available that depend on $T_w$, and one general source function used in Norwegian Earth System Model (NorESM),
Kirkevag et al., (2013) - $f_{kir}$ - which a modification of the Mårtensson et al., (2003) function adapted to the modal aerosol model in NoreSM by Struthers et al. (2011). In the Figure 9 we can see the relatively good comparisons of source functions with our measurement flux spectra in two wind speed classes.

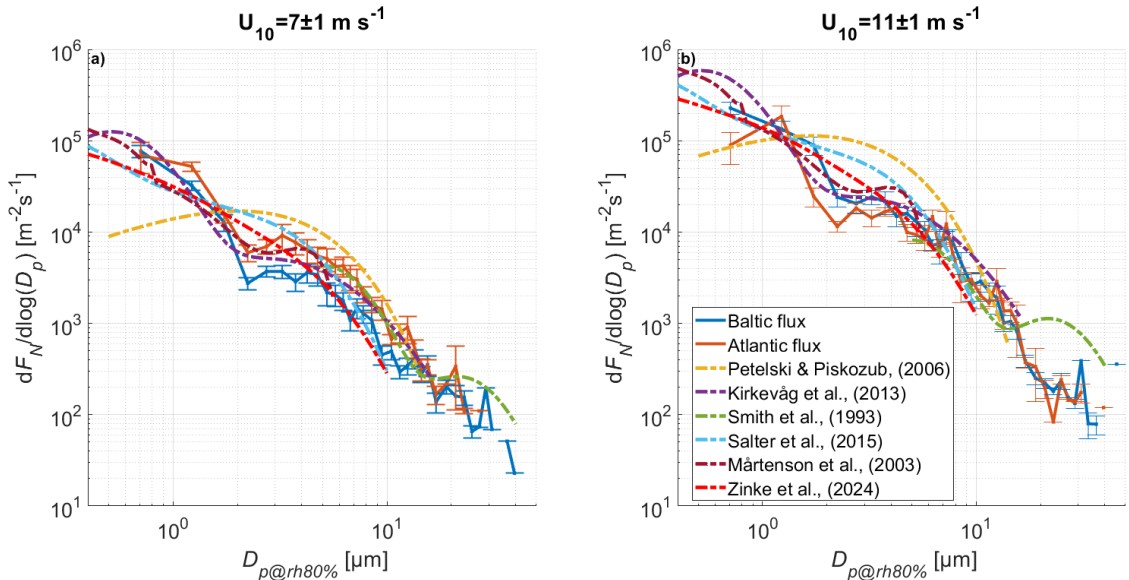

**Figure 9. Comparison of our measured sea spray aerosol number gradient fluxes size spectra with selected sea spray generation**
**functions related for two different wind speed cases (9a - 7 m s⁻¹, and 9b - 11 m s⁻¹). For the source parameterisations by Mårtensson et al. (2003), Struthers et al. (2011)/Kirkevåg et al. (2013) and Salter et al. (2015) used T$_w$=6.34°C.**





As evident in the Figure 9, the related $f_{kir}$ and $f_{mar}$ functions (originating from the same laboratory data) in particular reproduce our observations very well. The fit is decent for both sub micrometre particles and in the coarse particle range ($D_p > 2$ µm). It is also worth noting that the shape and location of the modal distribution agree with the results obtained. The $f_{sal}$ and $f_{zin}$

functions also reproduce the order of magnitude of SSA emission fluxes well. For these functions, a clear overestimation of the SSA emission fluxes in the aerosol size range from 1 to 4 µm $D_p$ is visible.

The $f_{sm}$ function, which is the oldest of the selected parameterizations dedicated to the largest aerosol particles (valid for $D_{p@rh80}$ > 6 µm - see Appendix A of de Leeuw et al., 2011), reproduces the emission well in the range from 5 µm to 15 µm. For larger particles, the function predicts the existence of another aerosol mode associated with spume droplets. We did not observe the

predicted increase in emission in this range. This is due to the fact that in order to perform accurate emission measurements in this size range, longer measurement times are necessary, to collect enough of the spume droplets which has a very small atmospheric number concentration. This is beyond the scope of our study.

We further compared the dependence of aerosol fluxes on the wave Reynolds number observed in this study to parameterizations by Ovadnevaite et al. (2014) and Zinke et al. (2024) assigned as $f_{ova}$ and $f_{zinRe}$ (Fig. 10).

As can be seen in the Figure 10, the $f_{ova}$ function underestimates the aerosol emission, especially for the submicron mode ($D_p<2$ µm). With increasing $ReH_w$ values (Fig. 10b), the $f_{ova}$ function much better reproduces the emission for larger particles ($D_p>1.5$ µm). In the case of the $f_{zinRe}$ function, we observe a good agreement with the Atlantic fluxes. The differences between the source functions and the measurements in the range from ~1.7 µm to ~3.2 µm are largely due to the different modal characteristics of the fits.

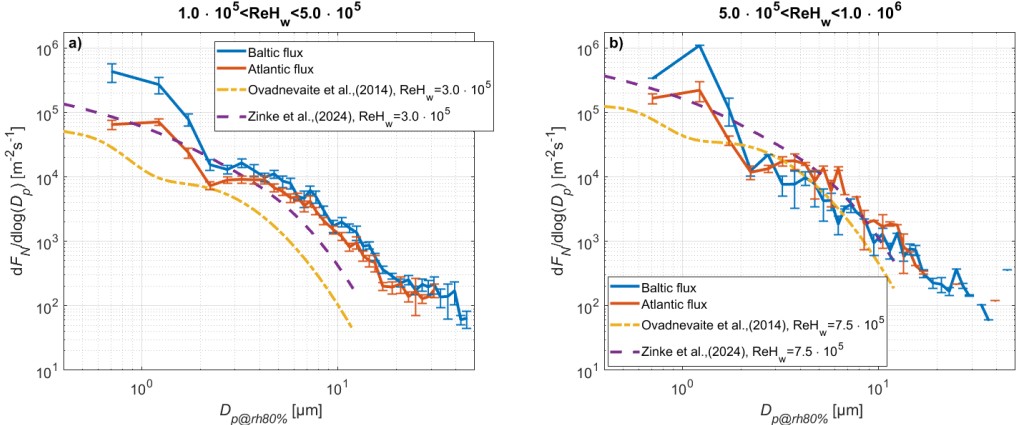


**Figure 10. Comparison of measured sea spray aerosol number gradient fluxes size spectra as function of $D_p$ (at 80 % relative humidity), with selected sea spray generation function related to two different wave Reynolds number cases in Fig. 10a and 10b.**

In Figure 11 we compare the chosen source functions with our measurement data as a function of $U_{10}$, $H_s$ and $ReH_w$. The source functions were integrated over the size distribution range of this work (0.5 µm < $D_p$ < 47 µm).

For $U_{10}$ < 8 m s$^{-1}$, the $P_A(U_{10})$ function (the Atlantic emissions) shows a good agreement with the other source functions. For larger wind speeds, the discrepancy increases due to the different slope of the curve compared to the other models. The $P_B^\beta$



function (low Chl-a concentrations) is characterized by a similar slope to the compared literature functions, but shows slightly lower values than the preexisting functions over the entire $U_{10}$ range. In the case of the Baltic fit $P_B^\alpha$ (all Baltic data), we observe significantly lower values than what the other source functions predict. The slope of this function is similar to the fit

to the Atlantic fluxes, but these are at a higher magnitude for the same $U_{10}$.

In the case of the functional dependence of aerosol emission on significant wave height (Fig. 11b), we observe large discrepancies. The fit of the function for the Atlantic measurements is clearly lower than the $F_{zinHs}$ function. The Baltic fit, on the other hand, is clearly higher than the predicted emissions, which suggests that $F_{zinHs}$ function does not reach all the way in its attempt to represent the Baltic Sea. We speculate the cause of this is different region of measurements. The function $F_{zinHs}$

were made based on experiments in the Baltic Proper/East Gotland Basin while this research in the Southern Baltic area.

In comparison, the $f_{ova}$ function is clearly lower compared to both our new functions based on gradient fluxes and the $f_{zinRe}$ function. The $f_{zinRe}$ function for values of $ReH_w = 1.7 \cdot 10^5$ and $< 10^5$ m$^{-2}$ s$^{-1}$ SSA emissions, is similar in value to both our Baltic and Atlantic fits. For higher $ReH_w$ values, the functions deviate from each other, and we observe similar discrepancies as in the case of the $H_s$ dependence - the Baltic parameterization is higher, while the Atlantic parameterization is lower than the

$F_{zinRe}$ function. The $f_{ova}$ function against $ReH_w$ clearly underestimates the emission flux with almost an order of magnitude.

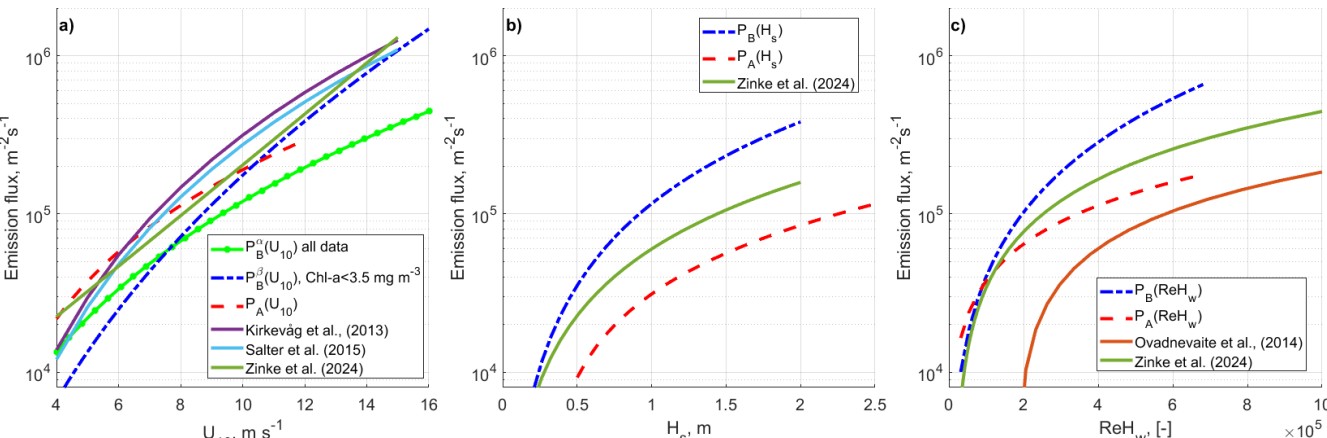

**Figure 11. Comparison between the parameterizations presented in this work with selected literature source functions: wind speed on panel a, significant wave height panel b, and Reynolds number on panel c.**

**4.4 The comparison of sea spray fluxes from the Baltic Sea and the Atlantic Ocean**

In order to compare the properties of the Baltic fluxes with the Atlantic fluxes, we again used the statistical tools used in section 4.1.1, namely the Welch t-test. The comparison was made in the wind speed range from 5 m s$^{-1}$ to 12 m s$^{-1}$ due to the comparable amount of data in both groups in these ranges. Both the aerosol number gradient fluxes and the aerosol volume gradient fluxes.

In the case of aerosol number fluxes, the results of statistical testing showed no difference between the Baltic fluxes measured

at low Chl-a concentrations and the Atlantic fluxes. However, the right-sided Welch test showed that aerosol fluxes for the entire Chl-a concentration range have significantly higher values than fluxes measured in the Atlantic. This result is due to the





observed increased Baltic aerosol emission in the 8-10 m s$^{-1}$ range for higher Chl-a concentrations visible in Figure 2. As shown in section 4.1.1, with further increase in wind speed, a statistically significant damping of Baltic aerosol emissions is observed compared to emissions measured at low Chl-a values. Due to the lack of observations of aerosol emission in the

Atlantic region at wind speeds greater than 12 m s$^{-1}$, the further evaluation of the phenomenon is not possible. In the case of volume fluxes, all tests showed no significant differences in the values of both data sets.

Based on our analysis, we speculate that the biological activity of seawater represented by Chl-a significantly contributes to the differences in SSA emissions. The observed lack of difference between volume fluxes suggests that this process mainly concerns smaller submicron particles, which dominates the aerosol number flux. This may be related to the reproducible result

from several studies that show that organic compounds contribute insignificantly to the super micrometre SSA, but in the sub micrometre, the organic sea spray fraction increase with decreasing diameter until it approaches 80 % around 0.1 μm $D_p$ (see references in the Introduction).

Another difference in emissions was observed when considering the influence of wave field properties. In order to compare statistically, we separated the data sets measured for "young" waves ($c_p/U_{10}<1$) and "old" waves ($c_p/U_{10}>1$). The result showed

that the emissions from the Baltic Sea for young waves are statistically significantly higher than the emissions from the Atlantic Ocean. In the case of old waves, the Welch test showed no statistically significant differences in aerosol emissions between the Atlantic Ocean and the Baltic Sea.

## 5 Summary and Conclusions

In this study, we present results of multi-year gradient measurements of aerosol fluxes in the Baltic Sea and the North Atlantic

Ocean from a research vessel. The amount of data collected allowed us to investigate the influence of a wide range of sea and atmosphere related parameters on total, and size distribution SSA fluxes in the range of 0.5 to 47 μm. We observed a statistically significant suppressing effect of marine biological matter represented by Chl-a concentration on SSA fluxes. The threshold Chl-a concentration value above which we observed the suppressing effect was 3.5 mg m$^{-3}$. Further parameterizations were performed for fluxes measured at Chl-a concentrations below this value in order to keep it comparable between the Baltic

and the Atlantic waters. We observed high and statistically significant correlations of SSA emissions with the following parameters: wind speed, wind acceleration, wave age, significant wave height, wave Reynolds number.

In addition, for sea water and air temperature, we observed, respectively: for the concentration flux - a negative correlation with temperatures, for the volume flux - a positive correlation. The values of both correlations and the presented parameterizations were statistically weak.

The main factors that we identified that influence the differences between the Baltic Sea and the Atlantic Ocean are the presence of Chl-a and the properties of the wave field. Statistical analysis showed that elevated Chl-a concentrations were associated with suppression of emissions at higher wind speeds (above 10 m s$^{-1}$) in the Baltic Sea. We also found no statistically significant differences between the Baltic Sea and the North Atlantic at low Chl-a concentrations. In the case of the wave field properties



represented by the wave age parameter, we showed that aerosol emissions from the Baltic Sea were statistically significantly
higher for "young" waves than from the Atlantic Ocean.

The parameterizations we developed were compared with other aerosol source functions. Our measured fluxes - both total
fluxes and size distributions - as a function of wind speed are in good agreement with previous studies. Our results are in
particularly good agreement with the parameterization of Kirkevåg et al., (2013). In the case of the variability of size
distributions as a function of Reynolds number, the parameterization of Zinke et al., (2024) well reproduces our measurements.
The Ovadnevaite et al., (2014) function, on the other hand, clearly underestimates aerosol emissions. For total fluxes dependent
from the significant wave height and wave Reynolds number Zinke et al. (2024) function overestimates Atlantic fluxes and
underestimates Baltic Sea fluxes.

Our study is the first to compare aerosol emission measurements using the same method in two different marine environments.
In addition, the analysis of a large database of aerosol fluxes measured under various atmospheric and oceanographic
conditions allows for a broader presentation of the influence of various factors on emission. The Baltic Sea is particularly
distinguished in this respect, where the dynamics of changes in the chemical composition of water and temperature changes
are greater than in the waters of the Atlantic Ocean. This work identifies possible factors influencing the differences between
these regions. There is still a more extensive sea spray aerosol observation effort needed. Especially the contribution of water
temperature, atmospheric stability and marine biology on sea spray fluxes should be investigated further.


*Data availability.* The data from this study is available at the IOPAN Geonetwork database (geonetwork.iopan.pl/) [DOI will
be inserted after the review].

*Author contributions.* PiM, PrzM and TP designed the experiments, PiM, PrzM, MK, INW, VD, TP carried out the field
experiments, The data analysis was conducted by PiM with the help of EDN, JZ, EMM, MS, DL, TP. PiM prepared the
manuscript with contribution of all co-authors.

*Competing interests.* The contact author has declared that none of the authors has any competing interests.

*Acknowledgements.* This research was supported by the Polish National Agency for Academic Exchange under the Bekker
Program (Decision PPN/BEK/2019/1/00043/DEC/1), through a National Science Centre grant (BaSEAf: Baltic Sea European
Arctic fluxes) id. number: 2015/17/N/ST10/02396, and the CROISSANT project financed by the Swedish Research Council,
contract 2018-04255.

We kindly thank the crew of the RV *Oceania* for all technical assistance and safety care during the cruise.



## Appendix A: The fluxes overview

**Table A1. The Baltic Sea flux overview.**

| No. | Date | duration | Lat. | Long. | Flux num. | Adv. type | Time over sea, h | fetch, km |
|---|---|---|---|---|---|---|---|---|
| 1 | 05-Apr-2011 | 05:30 | 55°30'N | 017°18'E | 11 | s | 7 | 250 |
| 2 | 07-Apr-2011 | 02:00 | 54°40'N | 017°48'E | 4 | l | - | 20 |
| 3 | 25-Oct-2011 | 03:30 | 54°18'N | 017°06'E | 7 | l | - | 5 |
| 4 | 28-Mar-2012 | 01:00 | 54°42'N | 018°42'E | 2 | s/l | 10 | 20 |
| 5 | 30-Mar-2012 | 01:30 | 54°42'N | 018°42'E | 3 | s/l | 11 | 5/300 |
| 6 | 01-Apr-2012 | 01:00 | 54°42'N | 018°42'E | 2 | l | - | 15 |
| 7 | 02-Apr-2012 | 01:30 | 54°42'N | 018°42'E | 3 | s/l | 10 | 15 |
| 8 | 10-Nov-2012 | 06:30 | 54°50'N | 018°20'E | 13 | l | - | 5 |
| 9 | 12-Nov-2012 | 07:00 | 54°52'N | 018°25'E | 14 | l | - | 5 |
| 10 | 15-Nov-2012 | 08:00 | 54°52'N | 018°25'E | 0 | l | - | 5 |
| 11 | 16-Nov-2012 | 09:30 | 55°35'N | 021°4'E | 14 | s/l | 6-18 | 150/500 |
| 12 | 21-Feb-2013 | 06:00 | 54°41'N | 018°46'E | 12 | l? | 6 | 165 |
| 13 | 22-Feb-2013 | 03:00 | 54°41'N | 018°46'E | 6 | l | 5 | 60 |
| 14 | 23-Feb-2013 | 12:00 | 54°41'N | 018°46'E | 24 | l | 4 | 50 |
| 15 | 28-Feb-2013 | 04:30 | 54°52'N | 018°22'E | 9 | s | 20 | 350 |
| 16 | 01-Mar-2013 | 06:00 | 54°42'N | 018°42'E | 12 | s/l | 10 | 5/250 |
| 17 | 14-Oct-2013 | 05:00 | 55°13'N | 016°41'E | 0 | s | 8 | 150 |
| 18 | 15-Oct-2013 | 03:30 | 55°13'N | 016°41'E | 0 | s | 6 | 60 |
| 19 | 17-Oct-2013 | 05:00 | 55°13'N | 016°41'E | 10 | s | 7 | 80 |
| 20 | 18-Oct-2013 | 09:30 | 55°13'N | 016°41'E | 20 | s | 6 | 110 |
| 21 | 31-Jan-2015 | 03:30 | 54°44'N | 019°08'E | 0 | l | - | 40 |
| 22 | 01-Feb-2015 | 01:00 | 54°44'N | 019°08'E | 0 | l | - | 40 |
| 23 | 07-Feb-2015 | 01:00 | 54°44'N | 019°08'E | 2 | l | - | 30 |
| 24 | 09-Feb-2015 | 02:00 | 54°39'N | 018°38'E | 4 | s | 10 | 263 |
| 25 | 22-Oct-2015 | 08:30 | 55°30'N | 017°17'E | 17 | s | 9 | 240 |
| 26 | 23-Oct-2015 | 13:00 | 55°30'N | 017°18'E | 26 | s | 7 | 130 |
| 27 | 31-Jan-2016 | 02:00 | 54°30'N | 018°40'E | 4 | l | - | 5 |




| 28 | 01-Feb-2016 | 06:00 | 54°30'N | 018°40'E | 12 | 1 | - | 5 |
| 29 | 02-Feb-2016 | 01:00 | 54°30'N | 018°40'E | 2 | 1 | - | 5 |
| 30 | 03-Feb-2016 | 06:00 | 54°30'N | 018°40'E | 12 | 1 | - | 5 |
| 31 | 04-Feb-2016 | 07:00 | 54°30'N | 018°40'E | 14 | 1 | - | 5 |
| 32 | 21-Apr-2016 | 11:30 | 55°02'N | 018°05'E | 23 | s | 10 | 240 |
| 33 | 22-Apr-2016 | 05:30 | 55°02'N | 018°05'E | 11 | s | 7 | 180 |
| 34 | 26-Apr-2016 | 12:30 | 55°02'N | 018°05'E | 25 | s | 16 | 110/200 |
| 35 | 28-Apr-2016 | 09:00 | 54°57'N | 016°23'E | 0 | s | 8 | 230 |
| 36 | 27-Oct-2016 | 04:30 | 54°25'N | 019°23'E | 9 | 1 | - | 3 |
| 37 | 28-Oct-2016 | 20:00 | 54°25'N | 019°23'E | 0 | 1 | - | 3 |
| 38 | 29-Oct-2016 | 06:30 | 54°35'N | 019°14'E | 13 | s | 6 | 260 |
| 39 | 30-Oct-2016 | 01:30 | 54°35'N | 019°14'E | 3 | s | 10 | 400 |
| 40 | 26-Jan-2017 | 11:30 | 55°34'N | 017°28'E | 17 | s | 7 | 170 |
| 41 | 27-Jan-2017 | 03:30 | 55°34'N | 017°28'E | 7 | s | 12 | 150 |

**Table A2. The Atlantic Ocean flux overview.**

| No. | date | duration | latitude | longitute | Flux num. |
|---|---|---|---|---|---|
| 1 | 23-Jun-2009 | 10:00 | 73°30' N | 013° 06' E | 6 |
| 2 | 25-Jun-2009 | 01:00 | 73° 30' N | 005° 12'E | 2 |
| 3 | 26-Jun-2009 | 02:00 | 75° 00' N | 005° 00' E | 4 |
| 4 | 01-Jul-2009 | 01:30 | 75° 48' N | 008°48' E | 3 |
| 5 | 03-Jul-2009 | 04:00 | 76° 30' N | 011° 00'E | 4 |
| 6 | 09-Jul-2009 | 18:30 | 77° 24' N | 013° 06' E | 9 |
| 7 | 22-Jun-2010 | 01:30 | 72° 30' N | 019° 18' E | 3 |
| 8 | 26-Jun-2010 | 00:00 | 75° 00' N | 006° 48' E | 2 |
| 9 | 27-Jun-2010 | 02:00 | 75° 00' N | 008° 30' E | 4 |
| 10 | 16-Jul-2012 | 05:00 | 78° 08' N | 008° 10' E | 6 |
| 11 | 27-Jun-2013 | 01:30 | 73° 30' N | 013° 30' E | 3 |
| 12 | 15-Jul-2013 | 01:00 | 78° 06' N | 005° 00' E | 2 |
| 13 | 16-Jul-2013 | 01:00 | 78° 00' N | 007° 00' E | 2 |
| 14 | 11-Jul-2014 | 06:30 | 76° 29' N | 005° 04' E | 5 |
| 15 | 22-Jun-2015 | 01:00 | 73° 29' N | 018° 04' E | 2 |
| 16 | 23-Jun-2015 | 15:00 | 73° 29' N | 013° 48' E | 12 |
| 17 | 01-Jul-2015 | 03:00 | 76° 29' N | 009° 02' E | 6 |



| 18 | 02-Jul-2015 | 01:30 | 76° 30' N | 013° 02' E | 3 |
|---|---|---|---|---|---|
| 19 | 08-Jul-2015 | 06:00 | 77° 31' N | 011° 22' E | 6 |
| 20 | 17-Jul-2015 | 01:30 | 80° 28' N | 012° 09' E | 3 |
| 21 | 21-Jul-2015 | 01:30 | 79° 24' N | 007°38' E | 3 |
| 22 | 02-Jul-2016 | 07:00 | 76° 30' N | 007° 57' E | 9 |
| 23 | 24-Jun-2017 | 03:00 | 75° 00' N | 015° 00' E | 4 |
| 24 | 26-Jun-2017 | 02:30 | 75° 00' N | 002° 00' E | 5 |
| 25 | 28-Jun-2017 | 01:30 | 73° 30' N | 007° 00' E | 3 |

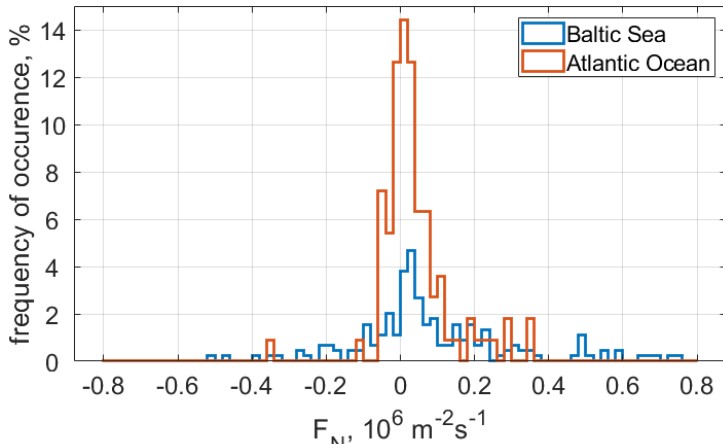

**Figure A1. Flux frequency distribution after the data cleaning from measurements on the Baltic Sea (blue line) and the Atlantic Ocean (orange line).**

**Appendix B: CSASP-100-HV description**

Description of Classical Scattering Aerosol Spectrometer PMS model CSASP-100-HV of Particle Measuring Systems inc.

The device was successfully used many times in different studies (inter alia: Hoppel et al. 1994, Jensen et al. 2001, Petelski, 2005, Savelyev et al. 2014, Markuszewski et al. 2017).

The CSASP uses a 5 mW He-Ne laser. The instrument samples an air volume of 12.6 cm$^3$ s$^{-1}$ at a flow velocity of 32.8 m s$^{-1}$. The device works in four size ranges mode. In each mode there are 15 channels of size bins. Mode 0: diameter size range 2-47 μm with interval 3 μm, mode 1: 2-32 μm with interval 2 μm, mode 2: 1-16 μm with 1 μm, and mode 3: 0.5-8.0 μm with size interval 0.5 μm.

The photo of the device and a way of placement are presented in the figure B1.



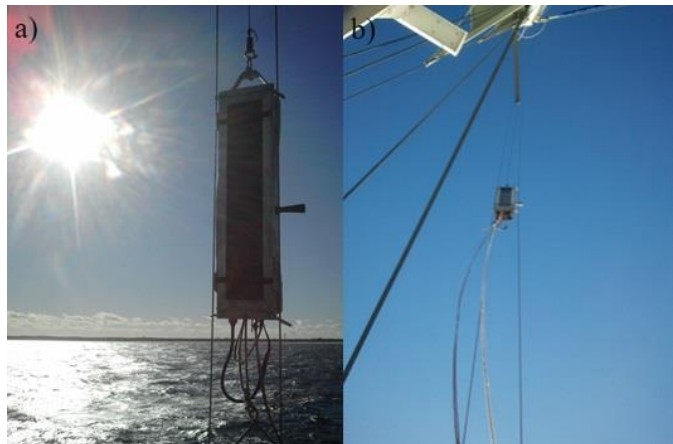

**Figure B1.** The classical aerosol spectrometer on the left panel and the method of device moving between altitude levels during gradient measurements.

**Appendix C: Sea spray aerosol fluxes calculated using an aerosol gradient**

To evaluate the aerosol gradient flux equation, let us start with the wind gradient formula:

$$\frac{dU}{dz} = \frac{u_*}{kz}\ ,\tag{C1}$$

Where *du/dz* is the wind speed profile of the function of altitude *z* and $u_*$ is the friction velocity defined from the definition of the momentum flux $\tau$ as:

$$\tau \equiv \rho u_*^2,\tag{C2}$$

Where $\rho$ is the fluid density.

The general solution of equation (C1) is given by Panofsky (1963):

$$u(z) = \frac{u_*}{kz}[\ln(z/z_0) - \psi_m(z/L)],\tag{C3}$$

Where $z_0$ is the roughness length for the wind profile, $\psi_m$ is the diabatic correction to the logarithmic wind profile in the Monin-Obuchov theory, and *L* is the Monin-Obukhov length.

The similar approach is used to other scalars like the potential temperature profile. Therefore, it is possible to also apply it for the aerosol concentration (*N*) profile:

$$\frac{dN(z,D_p)}{dD_p} = \frac{dN_*(D_p)}{dD_p}\ln(z) + C(D_p),\tag{C4}$$

where $N_*$ represents the aerosol concentration gradient analogues of $u_*$.

Using the least square method, where $N_*$ is linear coefficient of approximation and *C* is the free coefficient. Within aerosol
concentration gradient calculations the correlation coefficients *r* of each linear fitting were calculated. Results with correlations less than 0.4 were excluded from further analysis.

By differentiation of the above equation, we obtain:



$$\frac{d}{dz}\left(\frac{dN(D_p)}{dD_p}\right) = \frac{1}{z}\frac{dN_*(D_p)}{dD_p},$$ (C5)

Let's multiply both sides with momentum diffusivity coefficient with the assumption of near-neutral stratification ($K_h = kz\,u_*$)

and treat aerosol concentration as a scalar. Based on that, we can define aerosol flux:

$$-u_* kz\frac{d}{dz}\left[\frac{dN(D_p)}{dD_p}\right] = -ku_*\frac{dN_*(D_p)}{dD_p} \equiv \frac{dF_N(D_p)}{dD_p}.$$ (C6)

Using this formula, it is possible to determine aerosol vertical flux in near neutral stratification marine boundary layer. Detailed considerations about above formulas determinations are given by Panofsky (1963), Petelski (2003), and Andreas (2007).

Obtained aerosol concentration scales $N_*$ multiplied by $u_*$ gives aerosol concentration flux according to equation 6. $u_*$ was

calculated using our wind speed measurements using formula given by Large and Yeager (2004). This methodology was applied for total aerosol concentration and for each of 36 size bins.

Obtained fluxes were corrected for dry deposition flux, calculated in each size bin - using deposition velocity proposed by Schack et al. (1985).

At the end, all fluxes were also reduced to 80 % equilibrium humidity using Fitzgerald (1975) formula updated with the recent

sea spray growth factor given by Zieger et al. (2017). In this paper all aerosol and flux spectra are presented in relation to reduced diameters, which was marked later as $D_{p@80\%}$ - diameter at 80 % of humidity.

Exemplary results of such fitting are presented in the Fig. C1.

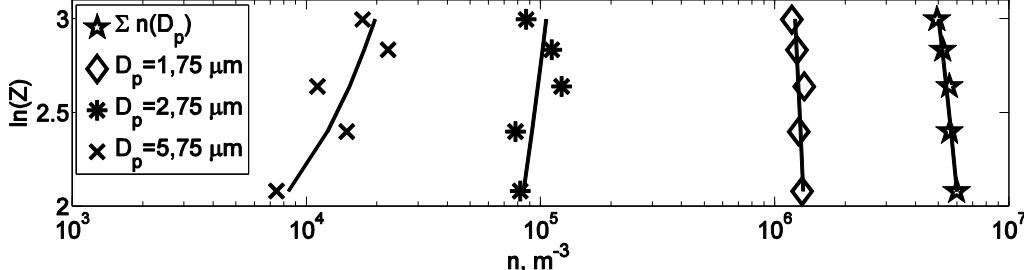

**Figure C1 Exemplary linear fitting to vertical profiles of aerosol concentration for exemplary size bins. Measurement carried out**
**during the Baltic Cruise, 21.02.2013.**

**Appendix D: Statistical tests results**

In order to proceed the statistical comparisons first we need to run the Levene's test. It is an inferential statistic used to assess the equality of variances for a variable calculated for two or more groups. Testing proved statistically significant difference between all analyzed pairs of data. After proved variance difference it is possible to run the Welch's *t*-test, (or unequal

variances *t*-test, Welch, 1947). It is a two-sample location test which is used to test the (null) hypothesis that two populations have equal means.



**Table D1. The Welch t-test results between source functions fitted to Baltic Sea fluxes. $P_B^\alpha(U_{10})$ represents function fitted to all data,**
**$P_B^\beta(U_{10})$ is the function fitted to low Chl-a cases only. The left side of the table shows test result for functions mean difference. On**
**the right side of the table the Right tailed Welch t-test results shows that for wind speed above 10 m s$^{-1}$ the function $P_B^\alpha(U_{10})$ is**
**significantly higher than function with low Chl-a case.**

| | General difference Welch t-test | | | | Right tailed Welch t-test | | | |
| | | | | | $U_{10}<10$ m s$^{-1}$ | | $U_{10}>10$ m s$^{-1}$ | |
| | $P_B^\alpha(U_{10})$ vs. $P_B^\beta(U_{10})$ | | $P_{vB}^\alpha(U_{10})$ vs. $P_{vB}^\beta(U_{10})$ | | $P_B^\alpha(U_{10})$ vs. $P_B^\beta(U_{10})$ | | $P_B^\alpha(U_{10})$ vs. $P_B^\beta(U_{10})$ | |
| | H | p-val | H | p-val | H | p-val | H | p-val |
|---|---|---|---|---|---|---|---|---|
| Test results | 1 | $5.8\cdot10^{-9}$ | 0 | 0.13 | 0 | 0.32 | 1 | $5.0\cdot10^{-4}$ |

**Table D2. Welch t-test results of total fluxes measured in the Baltic Sea region compared with fluxes measured in the Atlantic Ocean**
**area. Fluxes were compared for the same wind speed ranges of 5-12 m s$^{-1}$ (first two columns), and two cases of wave ages – young**
**$c_p/U_{10}<1$ and old $c_p/U_{10}>1$ (third and fourth columns).**

| | Baltic Chl-a <3.5 mg m$^{-3}$ vs. Atlantic (5-12 m s$^{-1}$) | | Baltic all Chl a vs. Atlantic (5-12 m s$^{-1}$) | | Baltic Chl-a <3.5 mg m$^{-3}$ vs. Atlantic, ($c_p/U_{10}<1$) | | Baltic Chl-a <3.5 mg m$^{-3}$ vs. Atlantic, ($c_p/U_{10}>1$) | |
| | H | p-val | H | p-val | H | p-val | H | p-val |
|---|---|---|---|---|---|---|---|---|
| Number flux | 0 | 0.42 | 1 | 0.02 | 1 | 0.04 | 0 | 0.17 |
| Volume flux | 0 | 0.48 | 0 | 0.31 | 0 | 0.82 | 0 | 0.93 |



**Table D3. Right tailed Welch t-test results of comparison between Baltic and Atlantic total fluxes in the same wind speed range (first column), and for young waves only (second column).**

| | Baltic all Chl-a vs. Atlantic 5 - 12 m s$^{-1}$ | | Baltic Chl-a < 3.5 mg m$^{-3}$ vs. Atlantic, $c_p/U_{10}<1$ | |
|---|---|---|---|---|
| | H | p-val | H | p-val |
| Number flux | 1 | 0.01 | 1 | 0.018 |

**Referecnes**

Abdalla, S., Janssen, P. A., & Bidlot, J. R. (2010). Jason-2 OGDR wind and wave products: Monitoring, validation and assimilation. *Marine Geodesy*, *33*(S1), 239-255. https://doi.org/10.1080/01490419.2010.487798

Aitken, J. (1890). On improvements in the apparatus for counting the dust particles in the
atmosphere. *Proceedings of the Royal Society of Edinburgh*, *16*, 135-172. https://doi.org/10.1017/S0370164600006222

Allen, S., Allen, D.,Moss, K., Le Roux, G., Phoenix, V.R., Sonke, J.E., 2020. Examination of the ocean as a source for atmospheric microplastics. PLoS ONE 15, 1–14. https://doi.org/10.1371/journal.pone.0232746

Alpert, P. A., Kilthau, W. P., Bothe, D. W., Radway, J. C., Aller, J. Y., & Knopf, D. A. (2015). The influence of
marine microbial activities on aerosol production: A laboratory mesocosm study. *Journal of Geophysical Research: Atmospheres*, *120*(17), 8841-8860. https://doi.org/10.1002/2015JD023469

Andreae, M. O., & Rosenfeld, D. J. E. S. R. (2008). Aerosol–cloud–precipitation interactions. Part 1. The nature and sources of cloud-active aerosols. *Earth-Science Reviews*, *89*(1-2), 13-41. https://doi.org/10.1016/j.earscirev.2008.03.001

Andreas, E. L. (2004). Spray stress revisited. *Journal of physical oceanography*, *34*(6), 1429-1440. https://doi.org/10.1175/1520-0485(2004)034<1429:SSR>2.0.CO;2

Andreas, E. L. (2007). Comment on "Vertical coarse aerosol fluxes in the atmospheric surface layer over the North Polar Waters of the Atlantic" by Tomasz Petelski and Jacek Piskozub. *Journal of Geophysical Research: Oceans*, *112*(C11). https://doi.org/10.1029/2007JC004184

Ault, A. P., Moffet, R. C., Baltrusaitis, J., Collins, D. B., Ruppel, M. J., Cuadra-Rodriguez, L. A., ... & Grassian, V. H. (2013). Size-dependent changes in sea spray aerosol composition and properties with different seawater conditions. *Environmental science & technology*, *47*(11), 5603-5612. https://doi.org/10.1021/es400416g

Barthelmeß, T., & Engel, A. (2022). How biogenic polymers control surfactant dynamics in the surface microlayer: insights from a coastal Baltic Sea study. *Biogeosciences*, *19*(20), 4965-4992.
https://doi.org/10.5194/bg-19-4965-2022

Bates, T. S., Anderson, T. L., Baynard, T., Bond, T., Boucher, O., Carmichael, G., ... & Wu, Y. (2006). Aerosol direct radiative effects over the northwest Atlantic, northwest Pacific, and North Indian Oceans: estimates based on in-situ chemical and optical measurements and chemical transport modeling. *Atmospheric Chemistry and Physics*, *6*(6), 1657-1732. https://doi.org/10.5194/acp-6-1657-2006





Bates, T. S., Quinn, P. K., Coffman, D. J., Johnson, J. E., Upchurch, L., Saliba, G., ... & Behrenfeld, M. J. (2020). Variability in marine plankton ecosystems are not observed in freshly emitted sea spray aerosol over the North Atlantic Ocean. *Geophysical Research Letters*, *47*(1), e2019GL085938. https://doi.org/10.1029/2019GL085938

Bowyer, P. A., Woolf, D. K., & Monahan, E. C. (1990). Temperature dependence of the charge and aerosol
production associated with a breaking wave in a whitecap simulation tank. *Journal of Geophysical Research: Oceans*, *95*(C4), 5313-5319. https://doi.org/10.1029/JC095iC04p05313

Callaghan, A., de Leeuw, G., Cohen, L., & O'Dowd, C. D. (2008). Relationship of oceanic whitecap coverage to wind speed and wind history. *Geophysical Research Letters*, *35*(23). https://doi.org/10.1029/2008GL036165

Carslaw, K. S., Boucher, O., Spracklen, D. V., Mann, G. W., Rae, J. G. L., Woodward, S., & Kulmala, M.
(2010). A review of natural aerosol interactions and feedbacks within the Earth system. Atmospheric Chemistry and Physics, 10(4), 1701-1737. https://doi.org/10.5194/acp-10-1701-2010

Cavalli, F., Facchini, M. C., Decesari, S., Mircea, M., Emblico, L., Fuzzi, S., ... & Dell'Acqua, A. (2004). Advances in characterization of size-resolved organic matter in marine aerosol over the North Atlantic. Journal of Geophysical Research: Atmospheres, 109(D24). https://doi.org/10.1029/2004JD005137

Chiliński, M. T., Markowicz, K. M., & Kubicki, M. (2018). UAS as a support for atmospheric aerosols research: case study. *Pure and Applied Geophysics*, *175*(9), 3325-3342. https://doi.org/10.1007/s00024-018-1767-3

Christiansen, S., Salter, M. E., Gorokhova, E., Nguyen, Q. T., & Bilde, M. (2019). Sea spray aerosol formation: Laboratory results on the role of air entrainment, water temperature, and phytoplankton biomass. *Environmental science & technology*, *53*(22), 13107-13116. https://doi.org/10.1021/acs.est.9b04078

Cochran, R. E., Ryder, O. S., Grassian, V. H., & Prather, K. A. (2017). Sea spray aerosol: The chemical link between the oceans, atmosphere, and climate. Accounts of chemical research, 50(3), 599-604. https://doi.org/10.1021/acs.accounts.6b00603

Darr, J., Gottuso, S., Alfarra, M., Birge, D., Ferris, K., Woods, D., … & Johnson, A. (2018). The hydropathy scale as a gauge of hygroscopicity in sub-micron sodium chloride-amino acid aerosols. The Journal of
Physical Chemistry A, 122(40), 8062-8070. https://doi.org/10.1021/acs.jpca.8b07119

De Leeuw, G., Andreas, E. L., Anguelova, M. D., Fairall, C. W., Lewis, E. R., O'Dowd, C., ... & Schwartz, S. E. (2011). Production flux of sea spray aerosol. *Reviews of Geophysics*, *49*(2). https://doi.org/10.1029/2010RG000349

Deike, L., Reichl, B.G., Paulot, F., 2022. A Mechanistic Sea Spray Generation Function Based on the Sea State
and the Physics of Bubble Bursting. AGU Advances 3. https://doi.org/10.1029/2022AV000750

Dror, T., Lehahn, Y., Altaratz, O., & Koren, I. (2018). Temporal-scale analysis of environmental controls on sea spray aerosol production over the South Pacific Gyre. *Geophysical Research Letters*, *45*(16), 8637-8646. https://doi.org/10.1029/2018GL078707

*E.U. Copernicus Marine Service Information;* https://doi.org/10.48670/moi-00014

Facchini, M. C., Rinaldi, M., Decesari, S., Carbone, C., Finessi, E., Mircea, M., ... & O'Dowd, C. D. (2008). Primary submicron marine aerosol dominated by insoluble organic colloids and aggregates. *Geophysical Research Letters*, *35*(17). https://doi.org/10.1029/2008GL034210

Ferrero, L., Castelli, M., Ferrini, B. S., Moscatelli, M., Perrone, M. G., Sangiorgi, G., ... & Cappelletti, D. (2014). Impact of black carbon aerosol over Italian basin valleys: high-resolution measurements along vertical
profiles, radiative forcing and heating rate. *Atmospheric Chemistry and Physics*, *14*(18), 9641-9664. https://doi.org/10.5194/acp-14-9641-2014

Ferrero, L., Scibetta, L., Markuszewski, P., Mazurkiewicz, M., Drozdowska, V., Makuch, P., ... & Bolzacchini, E. (2022). Airborne and marine microplastics from an oceanographic survey at the Baltic Sea: an emerging role of air-sea interaction?. *Science of the Total Environment*, *824*, 153709.
https://doi.org/10.1016/j.scitotenv.2022.153709




Fitzgerald, J. W. (1975). Approximation formulas for the equilibrium size of an aerosol particle as a function of its dry size and composition and the ambient relative humidity. Journal of Applied Meteorology, 14(6), 1044-1049. https://doi.org/10.1175/1520-0450(1975)014<1044:AFFTES>2.0.CO;2

Forestieri, S. D., Moore, K. A., Martinez Borrero, R., Wang, A., Stokes, M. D., & Cappa, C. D. (2018). Temperature and composition dependence of sea spray aerosol production. *Geophysical Research Letters*, *45*(14), 7218-7225. https://doi.org/10.1029/2018GL078193

Fuentes, E., Coe, H., Green, D., Leeuw, G.D., Mcfiggans, G., 2010. On the impacts of phytoplankton-derived organic matter on the properties of the primary marine aerosol – Part 1 : Source fluxes. Atmos. Chem. Phys. 10, 9295–9317. https://doi.org/10.5194/acp-10-9295-2010

Gantt, B., Meskhidze, N., Facchini, M. C., Rinaldi, M., Ceburnis, D., & O'Dowd, C. D. (2011). Wind speed dependent size-resolved parameterization for the organic mass fraction of sea spray aerosol. *Atmospheric Chemistry and Physics*, *11*(16), 8777-8790. https://doi.org/10.5194/acp-11-8777-2011

Garnesson P., A. Mangin, O. Fanton d'Andon, J. Demaria, and M. Bretagnon, Global Ocean Colour (Copernicus-GlobColour), Bio-Geo-Chemical, L3 (daily) from Satellite Observations (1997-ongoing), https://doi.org/10.48670/moi-00280

Geever, M., O'Dowd, C. D., van Ekeren, S., Flanagan, R., Nilsson, E. D., de Leeuw, G., & Rannik, Ü. (2005). Submicron sea spray fluxes. *Geophysical Research Letters*, *32*(15). https://doi.org/10.1029/2005GL023081

Grythe, H., Ström, J., Krejci, R., Quinn, P., & Stohl, A. (2014). A review of sea-spray aerosol source functions using a large global set of sea salt aerosol concentration measurements. *Atmospheric Chemistry and Physics*, *14*(3), 1277-1297. https://doi.org/10.5194/acp-14-1277-2014

Hanson, J. L., & Phillips, O. M. (1999). Wind sea growth and dissipation in the open ocean. *Journal of physical oceanography*, *29*(8), 1633-1648. https://doi.org/10.1175/1520-0485(1999)029<1633:WSGADI>2.0.CO;2

Hersbach, H., Bell, B., Berrisford, P., Hirahara, S., Horányi, A., Muñoz-Sabater, J., ... & Thépaut, J. N. (2020). The ERA5 global reanalysis. *Quarterly Journal of the Royal Meteorological Society*, *146*(730), 1999-2049. https://doi.org/10.1002/qj.3803

Hoppel, W. A. Frick, G. M. Fitzgerald, J. W. & Larson, R. E. (1994). Marine boundary layer measurements of new particle formation and the effects nonprecipitating clouds have on aerosol size distribution. Journal of Geophysical Research: Atmospheres, 99(D7), 14443-14459. https://doi.org/10.1029/94JD00797

Hultin, K. A., Krejci, R., Pinhassi, J., Gomez-Consarnau, L., Mårtensson, E. M., Hagström, Å., & Nilsson, E. D. (2011). Aerosol and bacterial emissions from Baltic Seawater. *Atmospheric research*, *99*(1), 1-14. https://doi.org/10.1016/j.atmosres.2010.08.018

Jaeglé, L., Quinn, P. K., Bates, T. S., Alexander, B., & Lin, J. T. (2011). Global distribution of sea salt aerosols: new constraints from in situ and remote sensing observations. *Atmospheric Chemistry and Physics*, *11*(7), 3137-3157. https://doi.org/10.5194/acp-11-3137-2011

Janssen, P. A., Abdalla, S., Hersbach, H., & Bidlot, J. R. (2007). Error estimation of buoy, satellite, and model wave height data. *Journal of Atmospheric and Oceanic Technology*, *24*(9), 1665-1677. https://doi.org/10.1175/JTECH2069.1

Jensen, D. R. Gathman, S. G. Zeisse, C. R. McGrath, C. P. de Leeuw, G. Smith, M. A. ... & Davidson, K. L. (2001). Electro-optical propagation assessment in coastal environments (EOPACE): summaryand accomplishments. Optical Engineering, 40(8), 1486-1499. https://doi.org/10.1117/1.1387985

Keene, W. C., Maring, H., Maben, J. R., Kieber, D. J., Pszenny, A. A., Dahl, E. E., ... & Sander, R. (2007). Chemical and physical characteristics of nascent aerosols produced by bursting bubbles at a model air-sea interface. *Journal of Geophysical Research: Atmospheres*, *112*(D21). https://doi.org/10.1029/2007JD008464

Kirkevåg, A., Iversen, T., Seland, Ø., Hoose, C., Kristjánsson, J. E., Struthers, H., ... & Schulz, M. (2013). Aerosol–climate interactions in the norwegian earth system model–NorESM1-M. *Geoscientific Model Development*, *6*(1), 207-244. https://doi.org/10.5194/acp-15-11047-2015



Konik, M., Kowalewski, M., Bradtke, K., & Darecki, M. (2019). The operational method of filling information gaps in satellite imagery using numerical models. *International Journal of Applied Earth Observation and Geoinformation*, *75*, 68-82. https://doi.org/10.1016/j.jag.2018.09.002

Landwehr, S., O'Sullivan, N., Ward, B., 2015. Direct flux measurements from mobile platforms at sea: Motion and airflow distortion corrections revisited. Journal of Atmospheric and Oceanic Technology 32, 1163–1178. https://doi.org/10.1175/JTECH-D-14-00137.1

Large, W. G., & Yeager, S. G. (2004). Diurnal to decadal global forcing for ocean and sea-ice models: The data sets and flux climatologies. http://dx.doi.org/10.5065/D6KK98Q6

Lehahn, Y., Koren, I., Rudich, Y., Bidle, K. D., Trainic, M., Flores, J. M., ... & Vardi, A. (2014). Decoupling atmospheric and oceanic factors affecting aerosol loading over a cluster of mesoscale North Atlantic eddies. *Geophysical research letters*, *41*(11), 4075-4081. https://doi.org/10.1002/2014GL059738

Leppäranta, M., & Myrberg, K. (2009). *Physical oceanography of the Baltic Sea*. Springer Science & Business Media.

Levene, H. (1960). Robust tests for equality of variances. Contributions to probability and statistics, 278-292.

Lewis, E. R., & Schwartz, S. E. (2004). *Sea salt aerosol production: mechanisms, methods, measurements, and models* (Vol. 152). American geophysical union.

Liu, S., Liu, C. C., Froyd, K. D., Schill, G. P., Murphy, D. M., Bui, T. P., ... & Gao, R. S. (2021). Sea spray aerosol concentration modulated by sea surface temperature. *Proceedings of the National Academy of Sciences*, *118*(9), e2020583118. https://doi.org/10.1073/pnas.2020583118

Long, M. S., Keene, W. C., Kieber, D. J., Erickson, D. J., & Maring, H. (2011). A sea-state based source function for size-and composition-resolved marine aerosol production. *Atmospheric Chemistry and Physics*, *11*(3), 1203-1216. https://doi.org/10.5194/acp-11-1203-2011

Losi, N., Markuszewski, P., Rigler, M., Gregorič, A., Močnik, G., Drozdowska, V., ... & Ferrero, L. (2023). Anthropic Settlements' Impact on the Light-Absorbing Aerosol Concentrations and Heating
Rate in the Arctic. *Atmosphere*, *14*(12), 1768. **https://doi.org/10.3390/atmos14121768**

Markuszewski, P., Klusek, Z., Nilsson, E. D., & Petelski, T. (2020). Observations on relations between marine aerosol fluxes and surface-generated noise in the southern Baltic Sea. *Oceanologia*, *62*(4), 413-427. https://doi.org/10.1016/j.oceano.2020.05.001

Markuszewski, P., Kosecki, S., & Petelski, T. (2017). Sea spray aerosol fluxes in the Baltic Sea region: Comparison of the WAM model with measurements. Estuarine, Coastal and Shelf Science, 195, 16-22. https://doi.org/10.1016/j.ecss.2016.10.007

Markuszewski, P., Kosecki, S., & Petelski, T. (2017). Sea spray aerosol fluxes in the Baltic Sea region: Comparison of the WAM model with measurements. *Estuarine, Coastal and Shelf Science*, *195*, 16-22.
https://doi.org/10.1016/j.ecss.2016.10.007

Mårtensson, E. M., Nilsson, E. D., de Leeuw, G., Cohen, L. H., & Hansson, H. C. (2003). Laboratory simulations and parameterization of the primary marine aerosol production. *Journal of Geophysical Research: Atmospheres*, *108*(D9). https://doi.org/10.1029/2002JD002263

Marx, S., Lavin, K., Hageman, K., Kamber, B., O'Loingsigh, T., & McTainsh, G. (2014). Trace elements and
metal pollution in aerosols at an alpine site, new zealand: sources, concentrations and implications. Atmospheric Environment, 82, 206-217. https://doi.org/10.1016/j.atmosenv.2013.10.019

Massel, S. R. (1996). *Ocean surface waves: their physics and prediction* (Vol. 11). World scientific.

Massel, S. R. (2007). *Ocean waves breaking and marine aerosol fluxes* (Vol. 38). Springer Science & Business Media.

Massel, S. R. (2010). Surface waves in deep and shallow waters. *Oceanologia*, *52*(1), 5-52.





Mehta, S., Ortiz-Suslow, D. G., Smith, A. W., & Haus, B. K. (2019). A laboratory investigation of spume generation in high winds for fresh and seawater. *Journal of Geophysical Research: Atmospheres*, *124*(21), 11297-11312. https://doi.org/10.1029/2019JD030928

Meskhidze, N., Petters, M. D., Tsigaridis, K., Bates, T., O'Dowd, C., Reid, J., ... & Zorn, S. R. (2013). Production mechanisms, number concentration, size distribution, chemical composition, and optical properties of sea spray aerosols. https://doi.org/10.1002/asl2.441

Miles, J. W. (1957). On the generation of surface waves by shear flows. *Journal of Fluid Mechanics*, *3*(2), 185-204. https://doi.org/10.1017/S0022112057000567

Monahan, E. C. (1986). The ocean as a source for atmospheric particles. In *The role of air-sea exchange in geochemical cycling* (pp. 129-163). Dordrecht: Springer Netherlands.

Monahan, E. C., & O'MUIRCHEARTAIGH, I. G. (1986). Whitecaps and the passive remote sensing of the ocean surface. *International Journal of Remote Sensing*, *7*(5), 627-642. https://doi.org/10.1080/01431168608954716

Monahan, E. C., Davidson, K. L., & Spiel, D. E. (1982). Whitecap aerosol productivity deduced from simulation tank measurements. *Journal of Geophysical Research: Oceans*, *87*(C11), 8898-8904. https://doi.org/10.1029/JC087iC11p08898

Monahan, E. C., Spiel, D. E., & Davidson, K. L. (1986). A model of marine aerosol generation via whitecaps and wave disruption. In *Oceanic whitecaps: And their role in air-sea exchange processes* (pp. 167-174). Dordrecht: Springer Netherlands. https://doi.org/10.1007/978-94-009-4668-2_16

Mostafa H. Sharqawy, John H. Lienhard V, and Syed M. Zubair, "Thermophysical properties of seawater: A review of existing correlations and data," Desalination and Water Treatment, Vol. 16, pp.354-380, April 2010. (PDF file which includes corrections through June 2017.)

Mulcahy, J. P., O'Dowd, C. D., Jennings, S. G., & Ceburnis, D. (2008). Significant enhancement of aerosol optical depth in marine air under high wind conditions. *Geophysical Research Letters*, *35*(16). https://doi.org/10.1029/2008GL034303

Myrhaug, D., & Holmedal, L. E. (2008). Effects of wave age and air stability on whitecap coverage. *Coastal engineering*, *55*(12), 959-966. https://doi.org/10.1016/j.coastaleng.2008.03.005

Nayar K.G., M.H. Sharqawy, L.D. Banchik, and J.H. Lienhard V, "Thermophysical properties of seawater: A review and new correlations that include pressure dependence," Desalination, Vol. 390, pp.1-24, 2016. doi:10.1016/j.desal.2016.02.024 (preprint)

Nayar, K. G., Sharqawy, M. H., & Banchik, L. D. (2016). Thermophysical properties of seawater: A review and new correlations that include pressure dependence. *Desalination*, *390*, 1-24. https://doi.org/10.1016/j.desal.2016.02.024

Nilsson, E. D., & Rannik, Ü. (2001). Turbulent aerosol fluxes over the Arctic Ocean: 1. Dry deposition over sea and pack ice. *Journal of Geophysical Research: Atmospheres*, *106*(D23), 32125-32137. https://doi.org/10.1029/2000JD900605

Nilsson, E. D., Hultin, K. A., Mårtensson, E. M., Markuszewski, P., Rosman, K., & Krejci, R. (2021). Baltic sea spray emissions: In situ eddy covariance fluxes vs. simulated tank sea spray. *Atmosphere*, *12*(2), 274. https://doi.org/10.3390/atmos12020274

Nilsson, E. D., Rannik, Ü., Swietlicki, E., Leck, C., Aalto, P. P., Zhou, J., & Norman, M. (2001). Turbulent aerosol fluxes over the Arctic Ocean: 2. Wind-driven sources from the sea. *Journal of Geophysical Research: Atmospheres*, *106*(D23), 32139-32154. https://doi.org/10.1029/2000JD900747

Norris, S. J., Brooks, I. M., de Leeuw, G., Smith, M. H., Moerman, M., & Lingard, J. J. N. (2008). Eddy covariance measurements of sea spray particles over the Atlantic Ocean. *Atmospheric Chemistry and Physics*, *8*(3), 555-563. https://doi.org/10.5194/acp-8-555-2008





Norris, S. J., Brooks, I. M., Hill, M. K., Brooks, B. J., Smith, M. H., & Sproson, D. A. (2012). Eddy covariance measurements of the sea spray aerosol flux over the open ocean. *Journal of Geophysical Research: Atmospheres*, *117*(D7). https://doi.org/10.1029/2011JD016549

Norris, S. J., Brooks, I. M., Moat, B. I., Yelland, M. J., De Leeuw, G., Pascal, R. W., & Brooks, B. (2013). Near-surface measurements of sea spray aerosol production over whitecaps in the open ocean. *Ocean science*, *9*(1),
133-145. https://doi.org/10.5194/os-9-133-2013

O'Dowd, C. D., Lowe, J. A., Smith, M. H., & Kaye, A. D. (1999). The relative importance of non-sea-salt sulphate and sea-salt aerosol to the marine cloud condensation nuclei population: An improved multi-component aerosol-cloud droplet parametrization. *Quarterly Journal of the Royal Meteorological Society*, *125*(556), 1295-1313. https://doi.org/10.1002/qj.1999.49712555610

Ovadnevaite, J., Manders, A., De Leeuw, G., Ceburnis, D., Monahan, C., Partanen, A. I., ... & O'Dowd, C. D. (2014). A sea spray aerosol flux parameterization encapsulating wave state. *Atmospheric Chemistry and Physics*, *14*(4), 1837-1852. https://doi.org/10.5194/acp-14-1837-2014

Parent, P., Laffon, C., Trillaud, V., Grauby, O., Ferry, D., Limoges, A., … & Piazzola, J. (2023). Physicochemical characterization of aerosols in the coastal zone: evidence of persistent carbon soot in the
marine atmospheric boundary layer (mabl) background. Atmosphere, 14(2), 291. https://doi.org/10.3390/atmos14020291

Partanen, A. I., Dunne, E. M., Bergman, T., Laakso, A., Kokkola, H., Ovadnevaite, J., ... & Korhonen, H. (2014). Global modelling of direct and indirect effects of sea spray aerosol using a source function encapsulating wave state. *Atmospheric Chemistry and Physics*, *14*(21), 11731-11752. https://doi.org/10.5194/acp-14-11731-
1005     2014

Petelski, T. (2003). Marine aerosol fluxes over open sea calculated from vertical concentration gradients. Journal of aerosol science, 34(3), 359-371. https://doi.org/10.1016/S0021-8502(02)00189-1

Petelski, T., & Piskozub, J. (2006). Vertical coarse aerosol fluxes in the atmospheric surface layer over the North Polar Waters of the Atlantic. *Journal of Geophysical Research: Oceans*, *111*(C6).
https://doi.org/10.1029/2005JC003295

Petelski, T., Markuszewski, P., Makuch, P., Jankowski, A., & Rozwadowska, A. (2014). Studies of vertical coarse aerosol fluxes in the boundary layer over the Baltic Sea. *Oceanologia*, *56*(4), 697-710. https://doi.org/10.5697/oc.56-4.697

Petelski, T., Piskozub, J., & Paplińska-Swerpel, B. (2005). Sea spray emission from the surface of the open Baltic
Sea. *Journal of Geophysical Research: Oceans*, *110*(C10). https://doi.org/10.1029/2004JC002800

Phillips, O. M. (1957). On the generation of waves by turbulent wind. *Journal of fluid mechanics*, *2*(5), 417-445. https://doi.org/10.1017/S0022112057000233

Podzimek, J. (1989). John Aitken's contribution to atmospheric and aerosol sciences—One hundred years of condensation nuclei counting. *Bulletin of the American Meteorological Society*, *70*(12), 1538-1545.
http://www.jstor.org/stable/26227760

Quinn, P. K., Bates, T. S., Schulz, K. S., Coffman, D. J., Frossard, A. A., Russell, L. M., ... & Kieber, D. J. (2014). Contribution of sea surface carbon pool to organic matter enrichment in sea spray aerosol. *Nature Geoscience*, *7*(3), 228-232. https://doi.org/10.1038/ngeo2092

Quinn, P. K., Coffman, D. J., Johnson, J. E., Upchurch, L. M., & Bates, T. S. (2017). Small fraction of
marine cloud condensation nuclei made up of sea spray aerosol. *Nature Geoscience*, *10*(9), 674-679.

Rak, D. (2016). The inflow in the Baltic Proper as recorded in January–February 2015. *Oceanologia*, *58*(3), 241-247.



Rap, A., Scott, C. E., Spracklen, D. V., Bellouin, N., Forster, P. M., Carslaw, K. S., ... & Mann, G. (2013). Natural aerosol direct and indirect radiative effects. *Geophysical Research Letters*, *40*(12), 3297-3301. https://doi.org/10.1002/grl.50441

Ribeiro, C.P., Mewes, D., 2006. On the effect of liquid temperature upon bubble coalescence. Chemical Engineering Science 61, 5704–5716. https://doi.org/10.1016/j.ces.2006.04.043

Salter, M. E., Nilsson, E. D., Butcher, A., & Bilde, M. (2014). On the seawater temperature dependence of the sea spray aerosol generated by a continuous plunging jet. *Journal of Geophysical Research: Atmospheres*, *119*(14), 9052-9072. https://doi.org/10.1002/2013JD021376

Salter, M. E., Zieger, P., Acosta Navarro, J. C., Grythe, H., Kirkevåg, A., Rosati, B., ... & Nilsson, E. D. (2015). An empirically derived inorganic sea spray source function incorporating sea surface temperature. *Atmospheric Chemistry and Physics*, *15*(19), 11047-11066. https://doi.org/10.5194/acp-15-11047-2015

Savelyev, I. B., Anguelova, M. D., Frick, G. M., Dowgiallo, D. J., Hwang, P. A., Caffrey, P. F., & Bobak, J. P. (2014). On direct passive microwave remote sensing of sea spray aerosol production. *Atmospheric Chemistry and Physics*, *14*(21), 11611-11631. https://doi.org/10.5194/acp-14-11611-2014

Schack Jr, C. J., Pratsinis, S. E., & Friedlander, S. K. (1985). A general correlation for deposition of suspended particles from turbulent gases to completely rough surfaces. *Atmospheric Environment (1967)*, *19*(6), 953-960. https://doi.org/10.1016/0004-6981(85)90240-9

Schwier, A. N., Sellegri, K., Mas, S., Charrière, B., Pey, J., Rose, C., ... & d'Anna, B. (2017). Primary marine aerosol physical flux and chemical composition during a nutrient enrichment experiment in mesocosms in the Mediterranean Sea. *Atmospheric Chemistry and Physics*, *17*(23), 14645-14660. https://doi.org/10.5194/acp-17-14645-2017

Seinfeld, J.H.; Pandis, S.N. 2006-Atmospheric Chemistry and Physics: From Air Pollution to Climate Change; J. Wiley: Hoboken, NJ, USA, 2006; ISBN 9780471720171.

Sellegri, K., Barthelmeß, T., Trueblood, J., Cristi, A., Freney, E., Rose, C., ... & Law, C. (2023). Quantified effect of seawater biogeochemistry on the temperature dependence of sea spray aerosol fluxes. *Atmospheric Chemistry and Physics*, *23*(20), 12949-12964. https://doi.org/10.5194/acp-23-12949-2023

Sellegri, K., Nicosia, A., Freney, E., Uitz, J., Thyssen, M., Grégori, G., ... & Law, C. S. (2021). Surface ocean microbiota determine cloud precursors. *Scientific reports*, *11*(1), 281. https://doi.org/10.1038/s41598-020-78097-5

Sha, B., Johansson, J. H., Benskin, J. P., Cousins, I. T., & Salter, M. E. (2020). Influence of water concentrations of perfluoroalkyl acids (PFAAs) on their size-resolved enrichment in nascent sea spray aerosols. Environmental Science & Technology, 55(14), 9489-9497. https://doi.org/10.1021/acs.est.0c03804

Sha, B., Johansson, J. H., Tunved, P., Bohlin-Nizzetto, P., Cousins, I. T., & Salter, M. E. (2021). Sea spray aerosol (SSA) as a source of perfluoroalkyl acids (PFAAs) to the atmosphere: field evidence from long-term air monitoring. *Environmental Science & Technology*, *56*(1), 228-238. https://doi.org/10.1021/acs.est.1c04277

Smith, M. H., Park, P. M., & Consterdine, I. E. (1993). Marine aerosol concentrations and estimated fluxes over the sea. *Quarterly Journal of the Royal Meteorological Society*, *119*(512), 809-824. https://doi.org/10.1002/qj.49711951211

Spiel, D.E., 1998. On the births of film drops from bubbles bursting on seawater surfaces. J . Geophys . Res . 103, 24907–24918. https://doi.org/10.1029/98JC02233

Stein, A.F., Draxler, R.R, Rolph, G.D., Stunder, B.J.B., Cohen, M.D., and Ngan, F., (2015). NOAA's HYSPLIT atmospheric transport and dispersion modeling system, Bull. Amer. Meteor. Soc., **96**, 2059-2077, http://dx.doi.org/10.1175/BAMS-D-14-00110.1

Stoń-Egiert, J., & Ostrowska, M. (2022). Long-term changes in phytoplankton pigment contents in the Baltic Sea: Trends and spatial variability during 20 years of investigations. *Continental Shelf Research*, *236*, 104666.





Stramska, M., & Petelski, T. (2003). Observations of oceanic whitecaps in the north polar waters of the Atlantic. *Journal of Geophysical Research: Oceans*, *108*(C3). https://doi.org/10.1029/2002JC001321

Struthers, H., Ekman, A., Glantz, P., Iversen, T., Kirkevåg, A., Mårtensson, E., ... & Nilsson, E. (2011). The effect of sea ice loss on sea salt aerosol concentrations and the radiative balance in the arctic. Atmospheric Chemistry and Physics, 11(7), 3459-3477. https://doi.org/10.5194/acp-11-3459-2011

Tsigaridis, K., Koch, D., & Menon, S. (2013). Uncertainties and importance of sea spray composition on aerosol direct and indirect effects. *Journal of Geophysical Research: Atmospheres*, *118*(1), 220-235. https://doi.org/10.1029/2012JD018165

Tyree, C. A., Hellion, V. M., Alexandrova, O. A., & Allen, J. O. (2007). Foam droplets generated from natural and artificial seawaters. *Journal of Geophysical Research: Atmospheres*, *112*(D12).
https://doi.org/10.1029/2006JD007729

Vaishya, A., Ovadnevaite, J., Bialek, J., Jennings, S. G., Ceburnis, D., & O'Dowd, C. D. (2013). Bistable effect of organic enrichment on sea spray radiative properties. *Geophysical research letters*, *40*(24), 6395-6398.

Veron, F. (2015). Ocean spray. *Annual Review of Fluid Mechanics*, *47*, 507-538. https://doi.org/10.1146/annurev-fluid-010814-014651

Walczowski, W., Beszczynska-Möller, A., Wieczorek, P., Merchel, M., & Grynczel, A. (2017). Oceanographic observations in the Nordic Sea and Fram Strait in 2016 under the IO PAN long-term monitoring program AREX. *Oceanologia*, *59*(2), 187-194. https://doi.org/10.1016/j.oceano.2016.12.003

Woolf, D. K. (2005). Parametrization of gas transfer velocities and sea-state-dependent wave breaking. *Tellus B: Chemical and Physical Meteorology*, *57*(2), 87-94. https://doi.org/10.3402/tellusb.v57i2.16783

Woolf, D. K., Bowyer, P. A., & Monahan, E. C. (1987). Discriminating between the film drops and jet drops produced by a simulated whitecap. *Journal of Geophysical Research: Oceans*, *92*(C5), 5142-5150. https://doi.org/10.1029/JC092iC05p05142

Woźniak, B., Bradtke, K., Darecki, M., Dera, J., Dudzińska-Nowak, J., Dzierzbicka-Głowacka, L., ... & Zapadka, T. (2011). SatBałtyk–A Baltic environmental satellite remote sensing system–an ongoing project in Poland.
Part 1: Assumptions, scope and operating range. *Oceanologia*, *53*(4), 897-924. https://doi.org/10.5697/oc.53-4.897

Woźniak, B., Bradtke, K., Darecki, M., Dera, J., Dudzińska-Nowak, J., Dzierzbicka-Głowacka, L., ... & Zapadka, T. (2011). SatBałtyk–A Baltic environmental satellite remote sensing system–an ongoing project in Poland. Part 2: Practical applicability and preliminary results. *Oceanologia*, *53*(4), 925-958.
https://doi.org/10.5697/oc.53-4.925

Xu, W., Ovadnevaite, J., Fossum, K. N., Lin, C., Huang, R. J., Ceburnis, D., & O'Dowd, C. (2022). Sea spray as an obscured source for marine cloud nuclei. Nature Geoscience, 15(4), 282-286. https://doi.org/10.17632/gjdd5r4ywf.1

Yang, X., Frey, M. M., Rhodes, R. H., Norris, S. J., Brooks, I. M., Anderson, P. S., ... & Wolff, E. W. (2019). Sea
salt aerosol production via sublimating wind-blown saline snow particles over sea ice: parameterizations and relevant microphysical mechanisms. *Atmospheric Chemistry and Physics*, *19*(13), 8407-8424. https://doi.org/10.5194/acp-19-8407-2019

Yoon, Y. J., Ceburnis, D., Cavalli, F., Jourdan, O., Putaud, J. P., Facchini, M. C., ... & O'Dowd, C. D. (2007). Seasonal characteristics of the physicochemical properties of North Atlantic marine atmospheric
aerosols. *Journal of Geophysical Research: Atmospheres*, *112*(D4). https://doi.org/10.1029/2005JD007044

Zábori, J., Matisāns, M., Krejci, R., Nilsson, E. D., & Ström, J. (2012). Artificial primary marine aerosol production: a laboratory study with varying water temperature, salinity, and succinic acid concentration. *Atmospheric Chemistry and Physics*, *12*(22), 10709-10724. https://doi.org/10.5194/acp-12-10709-2012



Zhao, D., & Toba, Y. (2001). Dependence of whitecap coverage on wind and wind-wave properties. *Journal of oceanography*, *57*, 603-616. https://doi.org/10.1023/A:1021215904955

Zieger, P., Väisänen, O., Corbin, J., Partridge, D., Bastelberger, S., Mousavi-Fard, M., … & Salter, M. (2017). Revising the hygroscopicity of inorganic sea salt particles. Nature Communications, 8(1). https://doi.org/10.1038/ncomms15883

Zinke, J., Nilsson, E. D., Markuszewski, P., Zieger, P., Mårtensson, E. M., Rutgersson, A., ... & Salter, M. E. (2024). Sea spray emissions from the Baltic Sea: comparison of aerosol eddy covariance fluxes and chamber-simulated sea spray emissions. *Atmospheric Chemistry and Physics*, *24*(3), 1895-1918. https://doi.org/10.5194/acp-24-1895-2024

Zinke, J., Nilsson, E. D., Zieger, P., & Salter, M. E. (2022). The effect of seawater salinity and seawater

temperature on sea salt aerosol production. *Journal of Geophysical Research: Atmospheres*, *127*(16), e2021JD036005. https://doi.org/10.1029/2021JD036005