# Peer review of "Multi-year gradient measurements of sea spray fluxes over the Baltic Sea and the North Atlantic Ocean"

_EGUsphere, 2024_

## Author Comment (AC2)

Dear Reviewer 2,

Thank you very much for your thorough and insightful review of our manuscript. We greatly appreciate your positive feedback and the valuable suggestions you have provided. Your comments highlight critical areas that require further clarification and detail, and we believe addressing these points will significantly strengthen our paper. Below, we provide our responses to your general and specific comments. We highlighted reviewer comments in blue.

Here are several comments to further strengthen this paper:

Please provide more information about aerosol measurements. It is the foundation of this study. Does the CSASP sample at a 10 s sampling rate? What is the detection limit of this instrument? What is the sizing and concentration accuracy of this instrument?

During our measurements the sampling rate of the instrument was set up to 10 s. The detection limit of the device according to the manual is 0.5 µm.

We have now included this information in the text in chapters 3.2 and 3.5.

We also corrected and extended our uncertainty calculations. Calculations were completed with the opc uncertainty vs. size distribution figure added into Appendix C fig. C2 presented below as fig 1.

[Figure]

Figure 1 Relation of measurement uncertainties with aerosol size distribution measured with the CSASP.

When studying the influential factors of SSA fluxes, I noticed that the author used the parameters from global atmospheric reanalysis. The resolution of those parameters is in 1-hour temporal resolution and a spatial resolution of 0.5*0.5 degrees. How do those resolutions match with the ship-based measurements? Is it too big spatially? What is the cruise speed of the ship? Are those parameters relevant to the ship measurements? If so, Do the reanalysis parameters cause any uncertainty in the correlation? There is a wind speed measurement on the ship. Why do authors use U10 in the correlation, not the ship measurements? I wonder if we will also find a weak correlation using ship-based sea surface and air temperatures.

During the measurement in the Baltic Sea the ship was at the anchor maintaining the same position all the time (so zero drift). During the Atlantic campaigns the ship was drifting, the maximum drift speed was 1 kt (in high wind conditions).

We decided to use modelled parameters in order to keep homogeneity of our data sets. Through such a long period of measurements in different areas of measurements due to technical and economic reasons it was impossible to keep the measurements on the same level of quality

(especially water temperature measurements). By using reanalyzed data we keep consequently to one data source with comparable properties. It is worth noticing that similar approach was used by other scientists (e.g. Ovadnevaite et al.(2014)).

We also tested correlations with wind speed dependent parameters and the overall results did not changed. For instance correlation between total flux data (all chll-a cases) and measured wind speed is 0.74, while correlation between fluxes and $U_{10}$ is 0.84 (correlations presented in the fig. 3. Also, overall trends are the same. For the temperature parameters for technical reasons we did not have available sea and water temperature measurements available within all campaigns.

To test the relation between modelled and measured wind speeds please check below fig 2 the comparison between $U_{10}$ and measured wind speed with the linear fit.

[Figure]

Figure 2 Comparison between modelled wind speed $U_{10}$ and measured wind speed $U_{measured}$

Specific comments:

 Abstract: Do the SSA fluxes have large standard deviation values? Is it caused by yearly differences or seasonal variations? After "we developed separate parameterizations and compared them with previous studies", maybe specify "What do we learn from it? Or major takeaway?"

The reason for large standard deviation values likely is from a combination of annual cycles, variations following from synoptical scale weather changes, the random nature of turbulence, and the presence of both opposing sources and sinks (aerosol deposition fluxes), resulting in a far from normal distributed data set with both positive and negative fluxes.

We also expanded our abstract. We erased the sentence: "Using these factors, we developed separate parameterizations and compared them with previous studies." And instead we added: "Our findings indicate that higher Chl-a concentrations are associated with reduced SSA fluxes at higher wind speeds in the Baltic Sea, while the influence of wave age showed higher aerosol emissions in the Baltic Sea for younger waves compared to the Atlantic Ocean. These insights underscore the complex interplay between biological activity and physical dynamics in regulating SSA emissions".

Section 3.3,  line 258, "A single measurement on a single level last at least 2 minutes." Does that mean the CSASP measures one size distribution in the 2 minutes? Later, in line 303,  the paper stated, "The single measurement was 10 s". Please clarify.

Thank you for catching this inconsistency. Sampling rate during the measurements was set up at 10 s. 2 min lasted measurement at 1 level of altitude, after 2 min the probe was moved to the next level

and kept for another 2 min. Average values for each 2 min period was formed from the 10 s data readings. We corrected this within the text.

Line 260, it said, " flux was determined … based on vertical aerosol gradient, friction, velocity, and stability". Please specify the friction velocity and stability source. Is it from the global reanalysis or in situ measurements?

We agree this sentence is unclear. We used modelled environmental parameters: friction velocity was calculated based on Large and Yeager (2004) formula (mentioned in line 745 in the Appendix C). In order to calculate friction velocity, we used reanalyzed $U_{10}$ data. We have removed this part from the manuscript because it could confuse a reader and this information is stated in the detailed method description in the Appendix.

Section 3.5, what are the aerosol concentrations and size distributions of the ship-based measurement? Do you see any seasonal variation in those data?

Our fluxes were calculated based on aerosol concentrations measured on five levels of altitude. We decided to skip aerosol concentration data presentation due to we focus on fluxes. As far as in the Baltic area we did not conduct regular measurements in all seasons (only one spring measurements, no measurements during summer) we are not able to conclusively show any pattern for aerosol concentration variability. On the figure 2 we present results of aerosol size distribution vs. wind speed.

[Figure]

Figure 2 Aerosol size distribution in the Baltic Sea and the North Atlantic Ocean in wind speed bins.

In line 304, n represents the aerosol counts, right? If so, the unit is wrong. In addition, the value of n is closely related to the instrument sampling rate, aerosol flow rate, and typical ambient sea salt aerosol concentration. Providing those data is critical for the community to understand the phenomena and this error propagation. If the typical total number concentration (between 0.5 – 47 micron aerosol) is around 10 #/cc, then for 2 mins sampling, with 12.6 cc/s sampling flowrate, we will get 10*12.6*2*60= 15120 particles. With 10 seconds of sampling, we will get 10*12.6*10=1260 particles.

Thank you very much for catching this. Basic counting time of the device was set up on 10 s. Within 2 minutes the device was sweeping between four modes of sizing 10 s each. Based on this, we calculated mean values and combined the modes into one size distribution.

We made this information more clear in the text of the manuscript.